# In vivo functional diversity of midbrain dopamine neurons within identified axonal projections

Navid Farassat[1], Kauê Machado Costa[1†], Strahinja Stojanovic[1], Stefan Albert[2], Lora Kovacheva[1], Josef Shin[1], Richard Egger[1], Mahalakshmi Somayaji[1‡], Sevil Duvarci[1], Gaby Schneider[2], Jochen Roeper[1]*

[1]Institute for Neurophysiology, Goethe University, Frankfurt, Germany; [2]Institute for Mathematics, Goethe University, Frankfurt, Germany

*For correspondence:
roeper@em.uni-frankfurt.de

Present address: †National Institute on Drug Abuse Intramural Research Program, National Institutes of Health, Baltimore, United States; ‡Department of Psychiatry, Columbia University, New York, United States

**Competing interests:** The authors declare that no competing interests exist.

**Abstract** Functional diversity of midbrain dopamine (DA) neurons ranges across multiple scales, from differences in intrinsic properties and connectivity to selective task engagement in behaving animals. Distinct in vitro biophysical features of DA neurons have been associated with different axonal projection targets. However, it is unknown how this translates to different firing patterns of projection-defined DA subpopulations in the intact brain. We combined retrograde tracing with single-unit recording and labelling in mouse brain to create an in vivo functional topography of the midbrain DA system. We identified differences in burst firing among DA neurons projecting to dorsolateral striatum. Bursting also differentiated DA neurons in the medial substantia nigra (SN) projecting either to dorsal or ventral striatum. We found differences in mean firing rates and pause durations among ventral tegmental area (VTA) DA neurons projecting to lateral or medial shell of nucleus accumbens. Our data establishes a high-resolution functional in vivo landscape of midbrain DA neurons.

DOI: https://doi.org/10.7554/eLife.48408.001

## Introduction

The functional properties and the emerging diversity of the midbrain dopamine (DA) system have been extensively studied across many different biological scales (*Schultz, 2015*; *Watabe-Uchida et al., 2017*). These differences range from the single-cell level, including diverging gene expression profiles (*Kramer et al., 2018*; *Nichterwitz et al., 2016*; *Poulin et al., 2018*; *Saunders et al., 2018a*; *Tiklová et al., 2019*), cellular and biophysical properties (*Evans et al., 2017*; *Lammel et al., 2008*; *Tarfa et al., 2017*), as well as neurotransmitter co-release (*Chuhma et al., 2018*; *Kim et al., 2015b*; *Tritsch et al., 2012*; *Zhang et al., 2015*), up to different neural circuit affiliations (*Beier et al., 2019*; *Beier et al., 2015*; *Lammel et al., 2012*; *Lerner et al., 2015*; *Menegas et al., 2015*; *Ogawa et al., 2014*; *Tian et al., 2016*; *Watabe-Uchida et al., 2012*) and selective task engagement of DA subpopulations in awake behaving rodents (*Dautan et al., 2016*; *de Jong et al., 2019*; *Duvarci et al., 2018*; *Gunaydin et al., 2014*; *Howe and Dombeck, 2016*; *Keiflin et al., 2019*; *Lammel et al., 2012*; *Matthews et al., 2016*; *Menegas et al., 2018*; *Menegas et al., 2017*; *Parker et al., 2016*; *Patriarchi et al., 2018*; *Saunders et al., 2018b*; *Vander Weele et al., 2018*; *Yang et al., 2018*) and non-human primates (*Kim et al., 2015a*; *Matsumoto and Hikosaka, 2009*; *Matsumoto and Takada, 2013*; *Ogasawara et al., 2018*). Moreover, additional methods complementing classical single unit recordings, including voltammetric and fluorometric measurements of DA release (*Dreyer et al., 2016*; *Hamid et al., 2016*; *Howe et al., 2013*; *Kishida et al., 2016*; *Lippert et al., 2019*; *Patriarchi et al., 2018*) and in vivo calcium imaging

of DA axons in distinct striatal areas (*Howe and Dombeck, 2016*) further support the notion of a multi-layered diversity of the DA system *at work* (*Berke, 2018*).

Rather than a single global computation, the midbrain DA system has therefore emerged as a parallel processor carrying out multiple functions (*Haber, 2014*; *Kegeles et al., 2010*). Support for this concept that distinct DA neuron subtypes carry out different computational processes arose from the mapping of their in vitro cellular properties in combination with retrograde tracing. This showed that projection-identified groups of DA neurons, including those that project to specific sub-regions of the striatum, have marked differences in intrinsic biophysical properties and input-output relationships (*Lammel et al., 2008*; *Tarfa et al., 2017*). However, this idea is somewhat challenged by virally-mediated single DA neuron labelling studies that depicted the extensive arborisation of individual axons often across distinct striatal territories (*Aransay et al., 2015*; *Matsuda et al., 2009*).

Distinct DA subpopulations are also differentially affected by pathological processes driving major brain disorders such as Parkinson disease (PD) (*Obeso et al., 2017*), schizophrenia (*McCutcheon et al., 2019*) or drug abuse (*Lüscher, 2016*). In PD, a ventrolateral SN DA population is most severely affected (*Damier et al., 1999*; *Fu et al., 2016*; *Gibb and Lees, 1991*; *Kordower et al., 2013*), which might be indicative for an innate and/or acquired differential vulnerability towards alpha-synuclein aggregation-mediated neurodegeneration (*Surmeier et al., 2017*). In schizophrenia, PET-studies in patients strongly suggested an early and selective dysregulation of a DA subpopulation projecting to associative, dorsomedial striatal regions (*Kegeles et al., 2010*; *McCutcheon et al., 2019*), but cellular substrates have remained elusive. In models of drug addiction, a hierarchical sequence of DA release in different striatal territories has been identified (*Willuhn et al., 2012*). In depression models, based on chronic social defeat, selective hyperactivity in mesolimbic-projecting DA subpopulations has been identified as a causal factor in depression-like behaviors using projection-selective optogenetic interventions (*Cao et al., 2010*; *Chaudhury et al., 2013*; *Tye et al., 2013*).

Therefore, there is increasing evidence that the classic triadic decomposition of the midbrain DA system (*Björklund and Dunnett, 2007*) into a dorsal mesostriatal, ventral mesolimbic, and cortical prefrontal arm might not be sufficiently granular to capture the functional and pathophysiological diversity of DA neurons. For example, the characteristic high-frequency 'burst' discharge of midbrain DA neurons in vivo is most commonly associated with positive reward prediction error signaling (*Cohen et al., 2012*; *Eshel et al., 2015*; *Schultz, 2015*) (*Stauffer et al., 2016*; *Steinberg et al., 2013*), but there is increasing evidence that – depending on the DA subtype – burst discharge plays also a role for other functions, including action control (*da Silva et al., 2018*; *Jin and Costa, 2010*) and the signaling of salience, novelty, anticipated fear or non-reward prediction errors, even in cells located in the same anatomical region (*Kim et al., 2015a*; *Matsumoto and Hikosaka, 2009*; *Menegas et al., 2018*; *Schiemann et al., 2012*). Nevertheless, the rapidly progressing dissection of DA controlled neuronal circuits using optogenetic and molecular tracing methods suggests this diversity in behavioral functions is matched to specific anatomical substrates (*Beier et al., 2019*; *Beier et al., 2015*; *Lammel et al., 2012*; *Lerner et al., 2015*; *Menegas et al., 2015*; *Ogawa et al., 2014*; *Tian et al., 2016*; *Watabe-Uchida et al., 2012*).

In order to help closing the current resolution gap between anatomy and physiology, it is therefore crucial to study the functional properties of DA neurons with identified axonal projections in the intact brain to better understand how intrinsic cellular differences and network affiliations jointly compute functional output. Accordingly, we carried out a systematic comparison of baseline in vivo electrical properties of anatomically and immunohistochemically verified midbrain DA neurons with identified axonal projections. Given our previous finding that even the innocuous labelling of DA neurons with GFP can affect their in vivo properties (*Schiemann et al., 2012*), we also aimed to carefully establish and control an experimental protocol for retrograde tracing that enabled us to register genuine differences of in vivo activity among DA neurons.

With this approach, we found a number of surprising results. First, we identified an unanticipated functional diversity among SN DA neurons, even among those sharing the same projections, targeting the dorsolateral striatum. However, this diversity followed a clear medial-to-lateral topography. Moreover, DA neurons with distinct axonal projections but overlapping distributions in the midbrain displayed different electrical properties regarding bursting, pausing, and even baseline mean frequencies. Only some of these identified in vivo differences were predicted based on previous studies of diverging intrinsic in vitro properties between distinct DA subpopulations (e.g. rebound delays

and pausing) (*Evans et al., 2017*; *Lammel et al., 2008*; *Tarfa et al., 2017*), while other features, like differences in mean firing rates were surprising and indicated that certain neuronal network properties might override intrinsic differences in excitability. Overall, our study provides a refined functional in vivo landscape of the midbrain DA system embedded in an anatomical context of identified axonal projections.

## Results

### Retrograde tracing with highly-diluted (0.002%) fluorogold preserves in vivo electrophysiological properties of nigrostriatal dopamine neurons

Our aim was to characterize the electrophysiological in vivo properties of midbrain DA neurons with defined axonal projections in adult male C57Bl/6N mice. We noticed that the application of a conventional retrograde tracing protocol (2% fluorogold (FG); 500 nl; 50 nl/min; *Liu et al., 2003*) in the dorsal striatum (DS; bregma: 0.86 mm, lateral: 2.0 mm, ventral: 2.8 mm) resulted in structural damage of nigrostriatal axons already one week after infusion (*Figure 1A*, column one, FG-0, note reduced TH expression). In contrast, we observed no apparent structural damage using a ten-fold FG-dilution (0.2%, *Figure 1A*, FG-1, 2nd column from left), but surprisingly failed to detect any stable in vivo spontaneously active DA neurons in the substantia nigra (SN), again one week after infusion (N = 3;~10 tracts per animal, intra-nigral speed of electrode advancement 60 μm/min; coverage 4–5 mm Z-depth from skull, data not shown). However, in control animals using the same number of tracts and electrode parameters, we routinely detected between 3–6 SN DA neurons per animal (N = 12, data not shown).

Thus, we carried out a systematic dilution series to define a working concentration of FG, which would still result in substantial coverage of retrogradely labelled nigrostriatal DA neurons but at the same time maintain their stable in vivo electrophysiological properties, similar to those observed in control animals. With this strategy, we identified 0.002% FG (FG-3) to be the lowest concentration where a significant proportion (ca. 20–30%) of SN DA neurons displayed detectable tracer levels based on intrinsic FG fluorescence (*Figure 1A*, FG-3, column four; *Figure 1—figure supplement 1A*, signal to noise 3:1). This rate of detection was further improved by using a FG-antibody (see Materials and methods) combined with a fluorescently-labelled secondary antibody (*Figure 1B*). The FG-antibody generated very low background in the ventral midbrain (contralateral side) resulting in an improved signal-to-noise ratio (see *Figure 1—figure supplement 1B*; Signal to noise 10:1). Detailed inspection of the cellular distribution of FG-immunoreactivity within DA neurons revealed selective immunostaining in cytoplasmic vesicles, most likely to be lysosomes (*Wessendorf, 1991*).

Next, we probed for the in vivo electrophysiological properties of FG-3 retrogradely labelled nigrostriatal DA neurons, one week after infusion, in comparison to those from control animals (*Figure 2*). All in vivo recorded neurons reported in this study were juxtacellularly labelled with neurobiotin to define their neurochemical nature (i.e. tyrosine hydroxylase (TH)-immunopositive = dopaminergic) and cellular position in the midbrain with the identification of their respective axonal projection targets, indicated by the FG signal in the labelled neuron. For a sensitive comparison between FG-3-labelled and control DA neurons, we focused on those in the lateral substantia nigra (lSN), given that they are the most vulnerable to cellular damage among midbrain DA neurons (*Surmeier et al., 2017*). *Figure 2* compares the spontaneous in vivo activity of a DA lSN neuron in a control animal (panels B1-B4) with one from a FG-3-labelled DA lSN neuron of a traced mouse (panels C1-C4). Panel C4 displays the neurobiotin and FG-signal demonstrating that this DA lSN neuron projected to the dorsolateral striatum (DLS; bregma: 0.86 mm, lateral: 2.2 mm, ventral: 2.6 mm) and - similarly to the control neuron (panel B4) - was located in the lSN at bregma −3.08 mm. Note that discharge frequencies and patterns as well as the single action potential waveforms are very similar between the control (panel B1-B3) and the FG-3-labelled DA neuron (panel C1-C3).

*Figure 3* summarizes the electrophysiological group data of retrogradely labelled DA lSN neurons compared to DA controls in the same regions. The completely overlapping cellular positions of these two groups are depicted in *Figure 3A*. The inset shows the representative FG-injection site in the DLS. In line with the examples shown in *Figure 2*, quantitative analysis of firing frequency (means ± SEMs; control: 3.51 ± 0.41 Hz, FG: 3.51 ± 0.4 Hz; panel 3B), regularity (CV; control: 62.92 ± 7.55%, FG: 58.84 ± 7.29%; panel 3B) and quantitative descriptors of ISI-distribution patterns (kurtosis

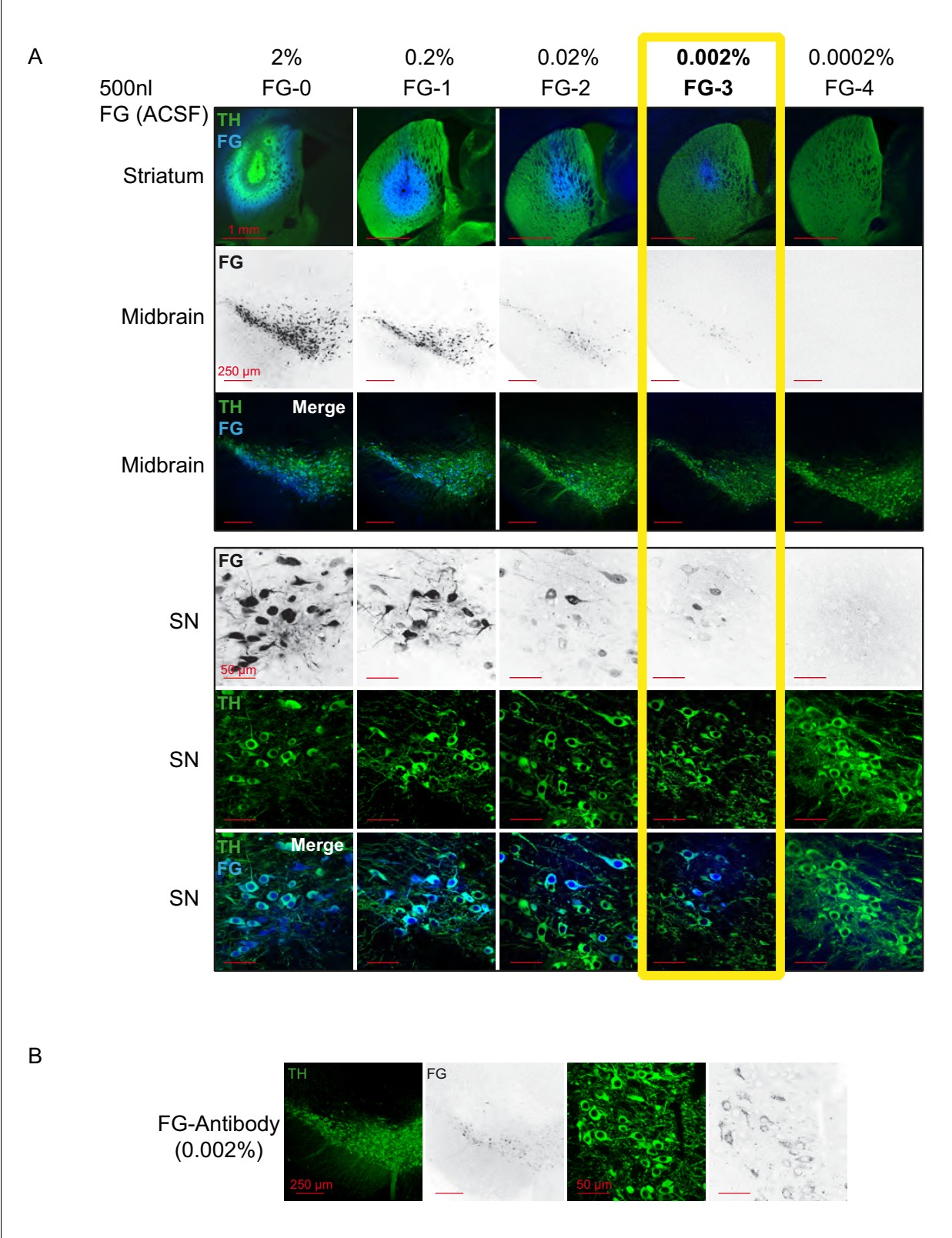

**Figure 1.** 1000-fold reduction of conventional FG-concentrations leads to intact FG-labelled DA neurons in vivo. (A) Confocal images of a dilution series of Fluorogold/ACSF injections. (Top panel, 1st row) merged images of injections sites (4x magnification) in dorsal striatum with TH in green and FG in blue. (2nd and 3rd row) 10x images of retrogradely traced neurons (FG black in the 2nd row and blue in the 3rd row) in the midbrain (TH green) at bregma −3.16 mm (10x magnification). (Bottom panel) Images of retrogradely traced SN DA neurons at higher magnification (60x; FG black in the 1st

*Figure 1 continued on next page*

*Figure 1 continued*
row and blue in the 3<sup>rd</sup> row, TH green). Note that 0.002% FG constitutes the lowest detectable concentration. (**B**) Use of FG-Antibody facilitates identification of FG-labelled neurons after 0.002% FG/ACSF infusion. Note the increased detectability in comparison to the intrinsic FG signal after 0.002% infusion of FG (*Figure 1A*, column four).
DOI: https://doi.org/10.7554/eLife.48408.002
The following figure supplement is available for figure 1:

**Figure supplement 1.** FG-labelling detection method.
DOI: https://doi.org/10.7554/eLife.48408.003

(control: 4.1 ± 0.73, FG: 3.69 ± 0.69), skewness (control: 0.98 ± 0.18, FG: 0.88 ± 0.16) and a firing pattern categorization based on a theoretical model called GLO (see Materials and methods), panel 3B) did not reveal any significant differences between the two populations. In addition, AP waveforms (AP duration (control: 2.04 ± 0.06 ms, FG: 1.94 ± 0.08 ms), relative AP trough (control: −0.63 ± 0.02, FG: −0.55 ± 0.04); panel 3C), burst and pause properties were also unaffected (SFB (control: 11.93 ± 4.18%, FG: 9.42 ± 3.57%), SFB contingency (% of neurons > 5% SFB; control: 37.5%, FG: 40%), bursts per minute (control: 7.67 ± 2.8, FG: 6.28 ± 2.24), intraburst-frequency (control: 21.37 ± 5.74 Hz, FG: 18.94 ± 1.72 Hz); panels 3E and 3F; pauses per minute (control: 0.93 ± 0.88, FG: 2.05 ± 1.26), pause duration (control: 0.88 ± 0.25 s, FG: 0.7 ± 0.13 s); panel 3D; see *Figure 3—figure supplement 1* for additional variables and their regional distributions). In summary, retrograde tracing with 1000-fold diluted FG did not perturb any in vivo electrophysiological property of DA lSN neurons. We assumed that less vulnerable DA populations residing in the medial SN and VTA are also likely to be unaffected by our FG labelling protocol. This in turn enabled us to compare authentic electrophysiological properties of DA midbrain subpopulations with defined axonal projections.

## Dorsal and ventral striatal-projecting DA neurons intermix in the medial substantia nigra

Based on our previous work (*Lammel et al., 2008*) that demonstrated axonal-projection-related differences in in vitro electrophysiological properties of midbrain DA neurons, we selectively traced axonal target areas in the dorsolateral (DLS; bregma: 0.86 mm, lateral: 2.2 mm, ventral: 2.6 mm) and dorsomedial striatum (DMS; bregma: 0.74 mm, lateral: 1.4 mm, ventral: 2.6 mm) as well as in the lateral (lNAcc; bregma: 0.86 mm, lateral: 1.75 mm, ventral: 4.5 mm) and medial shell (mNAcc; bregma: 1.54 mm, lateral: 0.7 mm, ventral: 4.25 mm) of the nucleus accumbens (*Figure 4A1–D1*; n = 3 animals each). One week after infusion, we mapped the respective striatal infusion sites (*Figure 4—figure supplement 3*) and the cellular distributions of TH+ (i.e. dopaminergic) neurons characterized by these four different projections (*Figure 4A2–D2*). The overall topographic patterns of cellular distributions (see *Figure 4—figure supplement 1*) was in accordance with those from previous work using retrobeads (*Lammel et al., 2008*) or retrograde viral tracers (*Lerner et al., 2015*). We also combined the tracing with additional calbindin immunohistochemistry, which confirmed the well-known calbindin-TH coexpression pattern but did not facilitate differentiating between different axonal projections (*Figure 4—figure supplement 2*).

Our data revealed a high complexity within the medial SN, where DA cell bodies of three distinct, equally prevalent axonal projections intermingled (mSN;~30% DLS,~40% DMS,~30% lNAcc). By contrast, in the two neighboring regions, the lSN and the lateral VTA (parabrachial pigmented nucleus (PBP), paranigral nucleus (PN)), only one axonal projection was dominant (~70% DLS-projecting DA neurons in the lSN;~60% lNAcc in the PBP,~70% mNAcc in the PN). These tracing results implied that just obtaining precise information on the cellular localization (SOURCE) of recorded DA neurons via juxtacellular labelling, in particular in the mSN, would be insufficient to capture the in vivo functional diversity of the DA system.

We then asked whether DA neurons from the same midbrain area (SOURCE) but with distinct axonal projections (TARGETS) displayed different in vivo electrophysiological properties (*Figure 4E2* – shared SOURCE and distinct TARGETS). We also compared DA neurons in the mSN with those in the lSN (SOURCES) that both projected to the same TARGET region, the dorsolateral striatum

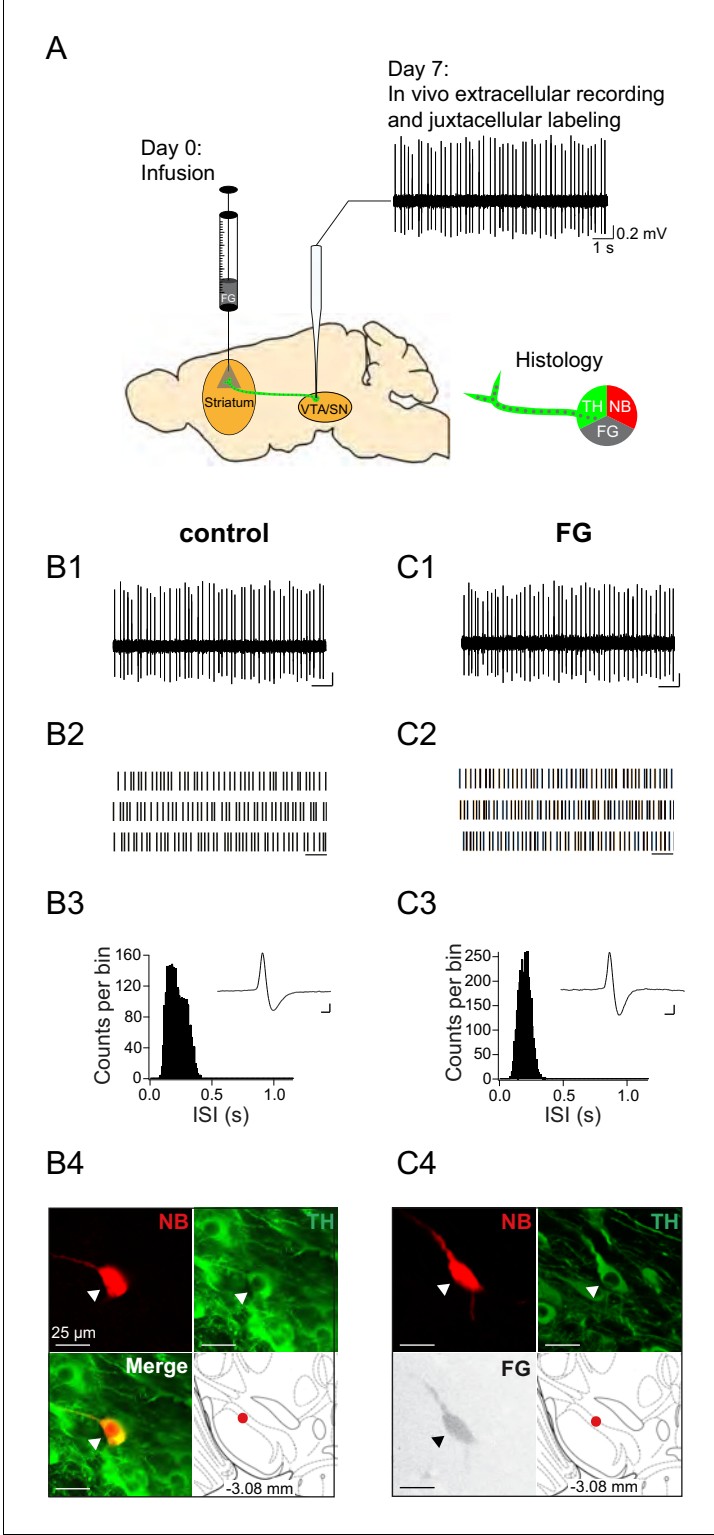

**Figure 2.** Experimental design and electrophysiological data of control neuron and FG-labelled SN DA neuron. (**A**) Experimental design for FG-labelled DA dataset. Control DA neurons were recorded in untreated mice (only 'Day 7'). (**B1, C1**) spontaneous in vivo extracellular single-unit activities of a representative control DA lSN neuron (**B**) and a representative FG-labelled DLS-projecting DA lSN neuron (**C**), shown as 10 s of original recording traces (scale bar: 0.2 mV, 1 s). (**B2, C2**) 30 s raster plots (scale bar: 1 s). (**B3, C3**) ISI-distributions. Inset, averaged AP waveform showing biphasic extracellular action potentials in high resolution (scale bar: 0.2 mV, 1 ms). (**B4, C4**)
*Figure 2 continued on next page*

*Figure 2 continued*

Confocal images of extracellularly recorded and juxtacellularly labelled neurons, the location of the neurons in the lSN at bregma −3.08 mm is shown in the bottom right image. Note the similarities in mean frequency, regularity, firing pattern, AP waveform and anatomical location.

DOI: https://doi.org/10.7554/eLife.48408.004

(*Figure 4F2*). Indeed, only the dorsolateral striatum, in contrast to dorsomedial and ventral striatum, received two equally prominent axonal projections from the mSN and the lSN (*Figure 4F2*).

In order to quantify the potential degree of overlap between neighboring axonal projections of the DA system, we carried out double-labelling retrograde tracing experiments combining fluoro-gold and red bead infusions at distinct striatal sites. In accordance to our previous study (*Lammel et al., 2008*), we found only a small percentage, varying between 4–14%, of double-labelled DA neurons (*Figure 5B*). This enabled us to meaningfully compare the in vivo physiological properties of largely parallel, non-overlapping DA projections. These recordings were carried out under well-controlled isoflurane anesthesia, which was continuously titrated to result in a spontaneous breathing rate between 1–2 Hz. This approach was very stable across animals (>70% of animals, the isoflurane concentration was 1.2% in 100% $O_2$, flow rate 350 ml/min) and the small variations in isoflurane concentrations in the range between 1–2% did not correlate with differences in firing frequencies or patterns in any of the studied DA projections (see *Figure 4—figure supplement 4A–C*). Moreover, during anesthesia, under the conditions stated above, oscillatory cortical activity, observed under lighter isoflurane anesthesia, was largely suppressed (see see *Figure 4—figure supplement 4G and H*). In addition, we found that firing frequencies and patterns of mSN/VTA were comparable to those in corresponding DA neurons recorded previously in freely moving, awake mice in the homecage (for detailed comparison, see *Figure 4—figure supplement 4D–F*). In essence, these controls enabled us to reliably identify meaningful, different functional in vivo properties between largely parallel and distinct DA projections.

## Low in vivo burstiness in medial compared to lateral DLS-projecting SN DA neurons

We recorded, labelled and identified DLS-projecting DA neurons in the mSN as well as in the lSN. *Figure 6* compares the electrophysiological properties of two DLS-projecting DA neurons localized in the lSN (*Figure 6A1–A4, C*) and mSN (*Figure 6B1–B4, C*), respectively. The original 10 s sample recording traces, the 30 s raster plots and the two ISI histograms indicate that the depicted DLS-lSN DA neuron discharged in a highly bursty pattern, which was completely absent in the regularly discharging DLS-mSN DA neuron. Importantly, while about 40% of DLS-projecting DA neurons in the lSN (n = 6 of 15) displayed this type of robust bursting (SFB >5%) throughout the recording period, none of the identified DLS-projecting DA neurons located in the mSN (n = 0 of 9) fired in this bursty manner (*Figure 6D*). Comparison of the cumulative probability distributions of burstiness (percentage of spikes fired in bursts; SFB%) between the two DLS-projecting SN DA populations confirmed their differences in firing pattern. In contrast, detailed analysis of other electrophysiological parameters between the two DLS-projecting SN DA neurons detected no further differences (see *Figure 6—figure supplement 1*). We then identified the best subset of electrophysiological parameters in a linear discriminant analysis (LDA) using half of the data from the two DLS-projecting SN DA populations. Using the variables burstiness (SFB), relative AP trough and precision of spiking ($\log(\sigma_2)$), we could accurately classify the remaining DLS-projecting SN DA neurons into a lateral and medial group with a predictive power of 79.2% (p<0.05). When using the entire data set to train the linear classifier, almost all (92.6%) of DLS-projecting SN DA neurons were classified correctly (*Figure 6E*). In summary, our data demonstrated that SN DA neurons projecting to the DLS possess distinct bursting properties in vivo depending on where they are located within the SN. We therefore identified a topographically organized (i.e. medial vs lateral) in vivo diversity among SN DA neurons sharing the same axonal projections to dorsolateral striatum.

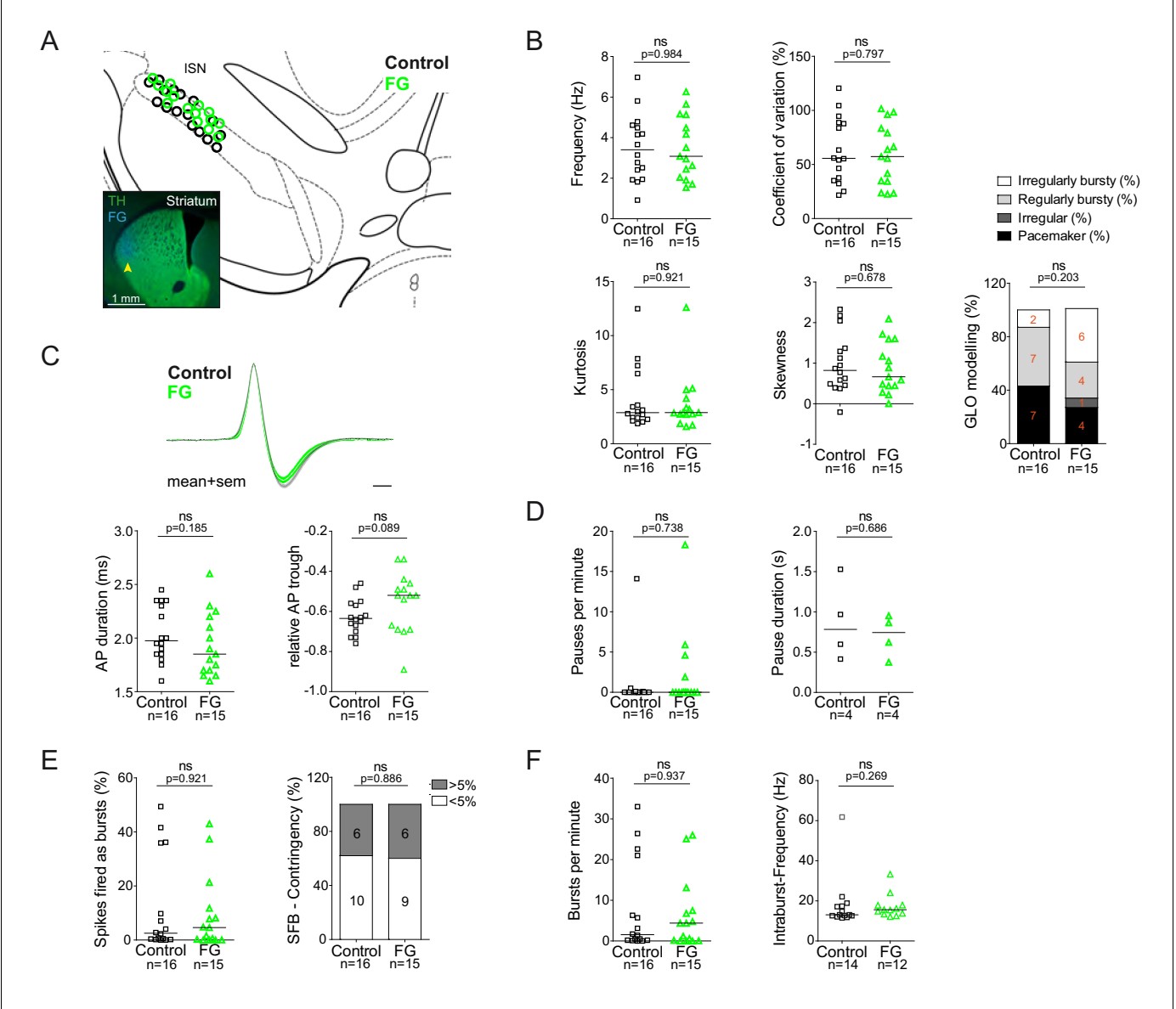

**Figure 3.** Retrograde tracing with highly-diluted (0.002%) fluorogold prevents perturbation of in vivo electrophysiological properties of identified nigrostriatal dopamine neurons. (A) Anatomical mapping of all extracellularly recorded and juxtacellularly labelled neurons (projected to bregma −3.16 mm; control in black, FG-labelled in green). Note the anatomical overlap of recorded DA populations in the lSN. Inset, FG-injection site in DLS (FG in blue, TH in green). (B–F) Scatter dot-plots (line at median) showing no significant differences in firing frequency (Hz), coefficient of variation (%), kurtosis and skewness of the ISI-distributions, GLO-based firing pattern (all B), normalized AP waveform, AP duration (ms), relative AP trough (all C), pauses per minute, pause duration (s) (both D), SFB (%), SFB contingency (% of neurons > and < 5% SFB) (both E), bursts per minute, intraburst-frequency (Hz) (both F).

DOI: https://doi.org/10.7554/eLife.48408.005

The following source data and figure supplement are available for figure 3:

**Source data 1.** Electrophysiological data.
DOI: https://doi.org/10.7554/eLife.48408.007
**Source data 2.** Labelling detection.
DOI: https://doi.org/10.7554/eLife.48408.008
**Figure supplement 1.** Comparison of in vivo firing properties of control and FG-labelled neurons.
DOI: https://doi.org/10.7554/eLife.48408.006

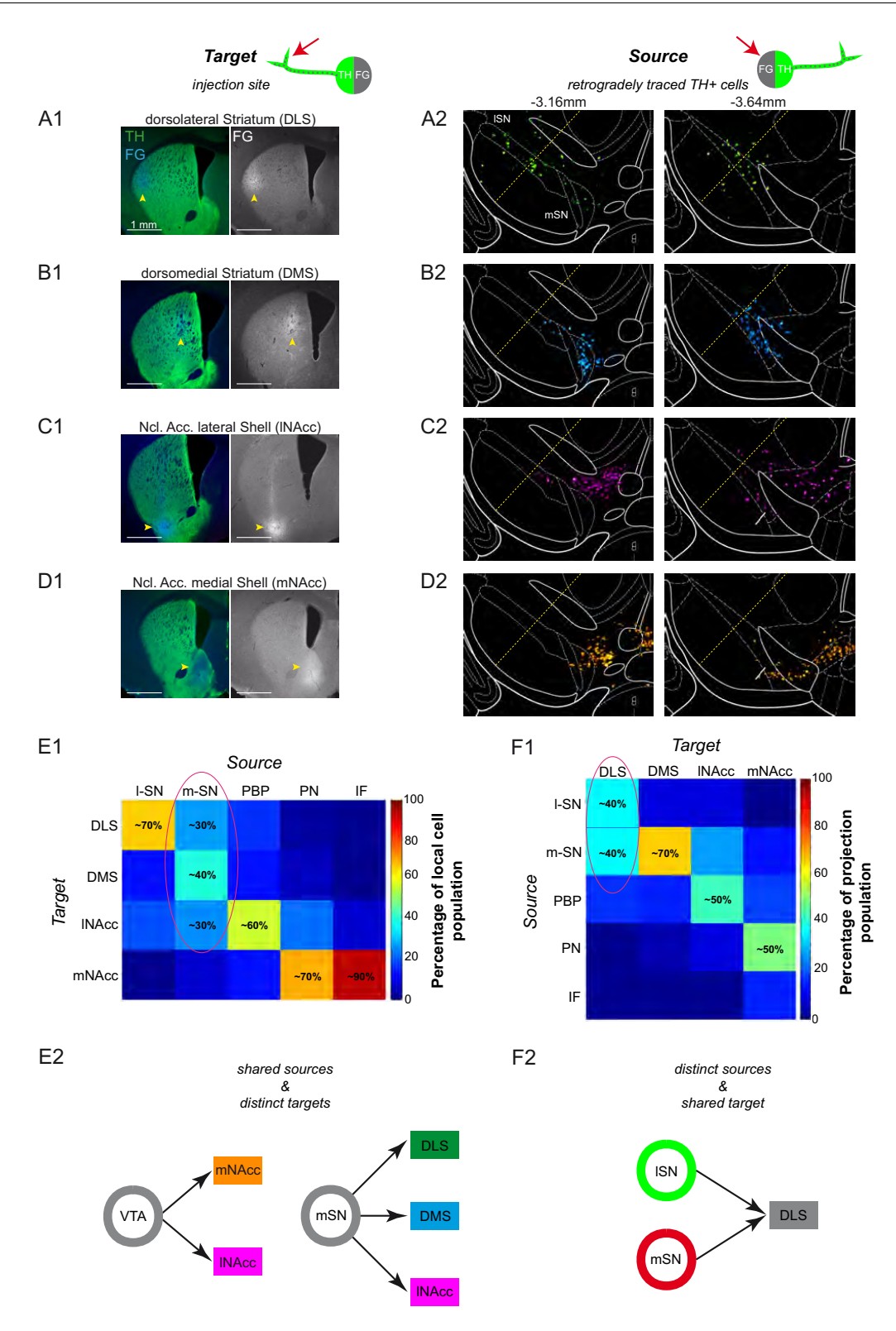

**Figure 4.** Dorsal and ventral striatal-projecting DA neurons intermix within the mSN. (**A1–D1**) Injection sites for fluorescent tracing experiments injecting FG into the DLS (A1; bregma +0.86 mm), DMS (B1; bregma +0.74 mm), lNAcc (C1; bregma +0.86 mm), mNacc (D1; bregma +1.54 mm). (Left images) merged TH-(green) and FG-(blue) signals. (Right images) FG mono. (**A2–D2**) locations of retrogradely traced DA neurons (accumulated from n = 3 animals) at a rostral (−3.16 mm) and a caudal (−3.64 mm) midbrain section, shown as colored heat maps. DLS-projecting DA neurons (**A2**) are

*Figure 4 continued*

shown in green, DMS-projecting DA neurons (**B2**) in cyan, lNAcc-projecting DA neurons (**C2**) in magenta, mNAcc-projecting DA neurons (**D2**) in orange. The border between lSN and mSN is indicated with a yellow dotted line. Note that DLS-projecting DA neurons are located throughout the mediolateral extent of the SN (**A2**) and that the mSN is a substantial midbrain source area for three mayor projection-defined systems (**A2–C2**). (**E1**) Local cell population matrix (%, average of three animals). Note that the lSN, PBP, PN and IF are populated mainly by one projection-defined population each (lSN - DLS, PBP - lNAcc, PN and IF - mNAcc). However, the mSN is a region of overlap of DLS-, DMS- and lNAcc- projecting DA neurons. (**F1**) Projection population matrix (%, average of three animals). Note that the DLS receives dopaminergic input from both SN source areas (lSN and mSN). On the other side, DMS mainly receives input from the mSN, lNAcc from PBP, mNAcc from PN. (**E2**) Shared sources and distinct targets. The mSN is a region of overlap of three different projection target-defined systems (DLS, DMS and lNAcc). The VTA harbors both the mNAcc and lNAcc projection system. (**F2**) Distinct sources and shared target. The DLS receives dopaminergic input from two distinct midbrain source areas (lSN and mSN).

DOI: https://doi.org/10.7554/eLife.48408.009

The following source data and figure supplements are available for figure 4:

**Source data 1.** Anatomical reconstruction.
DOI: https://doi.org/10.7554/eLife.48408.014
**Source data 2.** Anatomical reconstruction - DLS DLS-projecting cells counted in different regions and at distinct caudo-rostral levels.
DOI: https://doi.org/10.7554/eLife.48408.015
**Source data 3.** Anatomical reconstruction - DMS DMS-projecting cells counted in different regions and at distinct caudo-rostral levels.
DOI: https://doi.org/10.7554/eLife.48408.016
**Source data 4.** Anatomical reconstruction - lNAcc DLS-projecting cells counted in different regions and at distinct caudo-rostral levels.
DOI: https://doi.org/10.7554/eLife.48408.017
**Figure supplement 1.** Anatomical segregation of different DA systems in the midbrain.
DOI: https://doi.org/10.7554/eLife.48408.010
**Figure supplement 2.** Calbindin expression of midbrain DA neurons does not co-segregate with specific axonal projections.
DOI: https://doi.org/10.7554/eLife.48408.011
**Figure supplement 3.** Striatal fluorogold infusion sites show high reliability across animals.
DOI: https://doi.org/10.7554/eLife.48408.012
**Figure supplement 4.** In vivo activities of midbrain DA neurons recorded under anesthesia do not correlate with isoflurane concentrations.
DOI: https://doi.org/10.7554/eLife.48408.013
**Figure supplement 4—source data 1.** Electrophysiological data – DA neurons, recorded in awake mice, home cage.
DOI: https://doi.org/10.7554/eLife.48408.018

## Functional segregation of in vivo electrophysiological properties along distinct axonal projections of DA neurons in the medial substantia nigra

As described above (*Figure 4*), retrograde tracing revealed that the mSN harbors equal proportions of DA neurons projecting to dorsolateral, dorsomedial or ventral striatum (i.e. lateral shell of nucleus accumbens). We asked whether DA mSN neurons with distinct axonal projections also display systematic functional differences in vivo. We first compared the in vivo electrophysiology of mSN DA neurons projecting to DLS (n = 9) with those projecting to DMS (n = 9). Here, we found no differences between any parameters of these two mSN DA populations projecting to different territories of the dorsal striatum (*Figure 7—figure supplement 1*). Accordingly, the linear discriminant analysis (LDA) based on the best set of parameters (CV, SFB, skewness of the ISI-distribution, pauses per minute, the mean oscillation period (GLO) and the relative AP trough) did not perform above chance (50.9%) in discriminating DLS- from DMS-projecting DA neurons in the medial SN. We conclude that DA neurons localized in the medial SN (SOURCE) and projecting to different dorsal striatal areas (DMS/DLS, TARGETS) did not show significant differences in their in vivo electrophysiological baseline properties. Consequently, they were pooled (DS) for functional comparison with ventral projecting mSN DA neurons.

 *Figure 7* compares the electrophysiological properties of two DA neurons in the mSN, projecting either to dorsal striatum (DS) or ventral striatum (i.e. lateral shell of nucleus accumbens, lNAcc). The 10 s recording traces, 30 s raster plots and the two ISI histograms show obvious functional differences (panels 7A and 7B). Apart from mean frequencies and AP waveforms, which were similar, most electrophysiological parameters describing the patterns of discharge including bursts (*Figure 7D*) but also pauses (*Figure 7E*), as well as the shapes of the ISI-distributions (Kurtosis, Skewness; *Figure 7F*) were significantly different between dorsal (n = 18) and ventral (n = 14) striatum

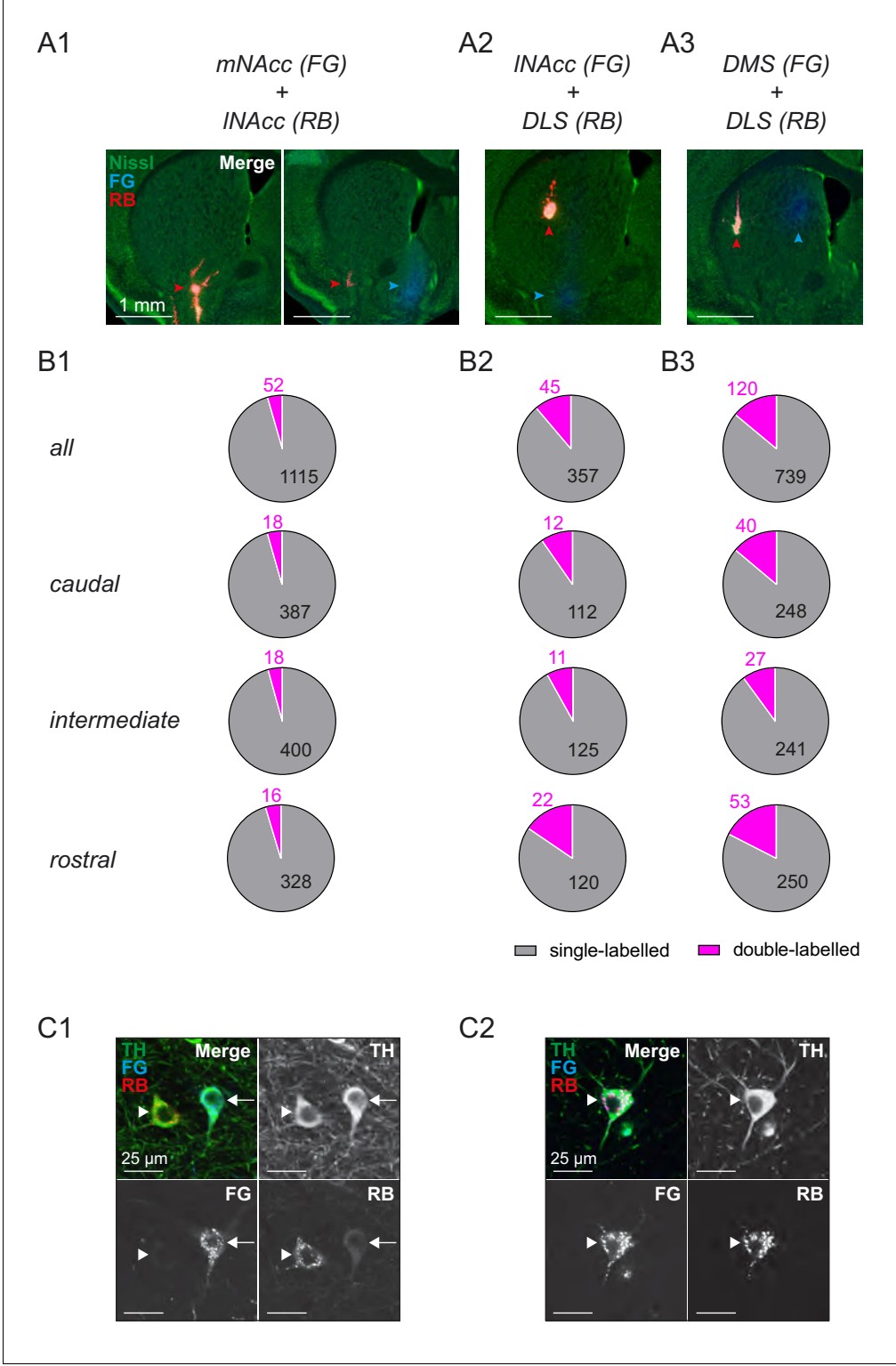

**Figure 5.** Projection-defined midbrain DA systems projecting to dorsal and ventral striatum are predominantly parallel and non-overlapping. (**A**) Infusion sites in green nissl counterstained 100 μm sections of double retrograde tracing experiments using fluorogold (blue) for mNAcc (**A1**), lNAcc (**A2**) and DMS (**A3**) and red fluorescently labelled latex beads (red) for lNAcc (**A1**) and DLS (**A2 and A3**). Arrowheads indicate the respective infusion sites. (**B**) Distribution of single-labelled and double-labelled DA neurons for each double tracing experiment (single-labelled in grey, double-labelled in magenta). Numbers indicate accumulated total numbers of

*Figure 5 continued on next page*

*Figure 5 continued*

neurons counted (first row, N = 3 per group), as well as those in caudal sections (−3.64 mm from bregma; second row), intermediate sections (−3.28 mm from bregma; third row) and rostral sections (−3.08 mm from bregma; fourth row). (C) Confocal images of TH-positive (green) cells in the m-SN labelled with fluorogold (blue) for lNAcc neurons and red beads (red) for DLS neurons. (C1) Example of two neighboring single-labelled neurons. (C2) Example of a double labelled neuron.

DOI: https://doi.org/10.7554/eLife.48408.019

The following source data is available for figure 5:

**Source data 1.** Double labelling Single-labelled and double-labelled.

DOI: https://doi.org/10.7554/eLife.48408.020

projecting mSN DA neurons (*Figure 7—figure supplement 2*). The linear discriminant analysis (LDA) based on the best set of parameters (skewness of the ISI-distribution, AP duration, relative AP trough and the precision of spiking (log(σ2)) showed a strong trend to classify mSN DA neurons into dorsal or ventral striatum projecting cells with a predictive power of 70.8%, when using half of the data set for training (p<0.07). When the entire data set was used, 84.4% of mSN DA neurons were classified correctly (*Figure 7G*). Furthermore, a functional comparison of lateral shell NAcc-projecting (TARGET) DA neurons localized in the mSN compared to those positioned in the VTA (SOURCES), the classical location of mesolimbic DA neurons, revealed no significant differences (see *Figure 7—figure supplement 3*) between the groups, violating the VTA/SN anatomical boundary. Therefore, lateral shell NAcc-projecting DA neurons from both mSN and VTA were pooled for further comparisons.

## DA neurons projecting to medial or lateral shell of the nucleus accumbens display distinct in vivo mean firing rates

We next asked whether distinct ventral striatal-projecting populations (i.e. lateral shell NAcc vs medial shell NAcc-projecting neurons) also show functional differences in vivo, given that these two populations displayed very distinct electrophysiological properties in vitro (*Lammel et al., 2008*). *Figure 8* compares the electrophysiological properties of two DA neurons in the VTA, projecting either to lateral shell (lNAcc) or medial shell (mNAcc) of nucleus accumbens. The 10 s recording traces, 30 s raster plots and the two ISI histograms display selective differences in mean discharge frequency and the duration of pauses only (*Figure 8A and B*), which are also evident at the population level (*Figure 8D*). In contrast, all other functional in vivo properties – including analyzed AP waveform parameters - did not differ between the two populations (see *Figure 8—figure supplement 1*). In light of the previous in vitro data, where medial shell NAcc-projecting DA neurons showed higher firing rates compared to lateral shell NAcc DA neurons, our in vivo results are surprising. In contrast, the longer in vivo pauses observed in medial shell NAcc projecting VTA DA neurons are in line with their very long in vitro post-inhibition rebound delays. Note that none of the subsets of variables used for linear discriminant analysis showed a segregation of these two mesolimbic DA populations above chance level.

Finally, we attempted to apply our approach to DA neurons projecting to medial prefrontal cortex (mPFC). While we succeeded in labelling and anatomically mapping this sparse DA subpopulation with our FG-3-protocol, the success rate of identifying and labelling spontaneously active mPFC-projecting DA neurons was not sufficient (ca. 10%) to carry out a systematic study (*Figure 8—figure supplement 2*; 1 of 10 recorded and labelled VTA neurons was identified to be dopaminergic and projecting to mPFC). As our method depends on spontaneous in vivo activity, prefrontal-projecting DA neuron might have to be targeted in awake animals.

## Discussion

Our study refines the functional in vivo topography of midbrain DA neurons in several ways. We provide the first detailed description and anatomical mapping of axon projection-specific functional DA phenotypes in vivo. Importantly, all n = 102 DA neurons in this study were labelled and immunohistochemically identified, thus avoiding the uncertainties of 'putative' DA populations. Moreover, our study controlled for functional perturbations by retrograde labelling itself, based on our high-dilution

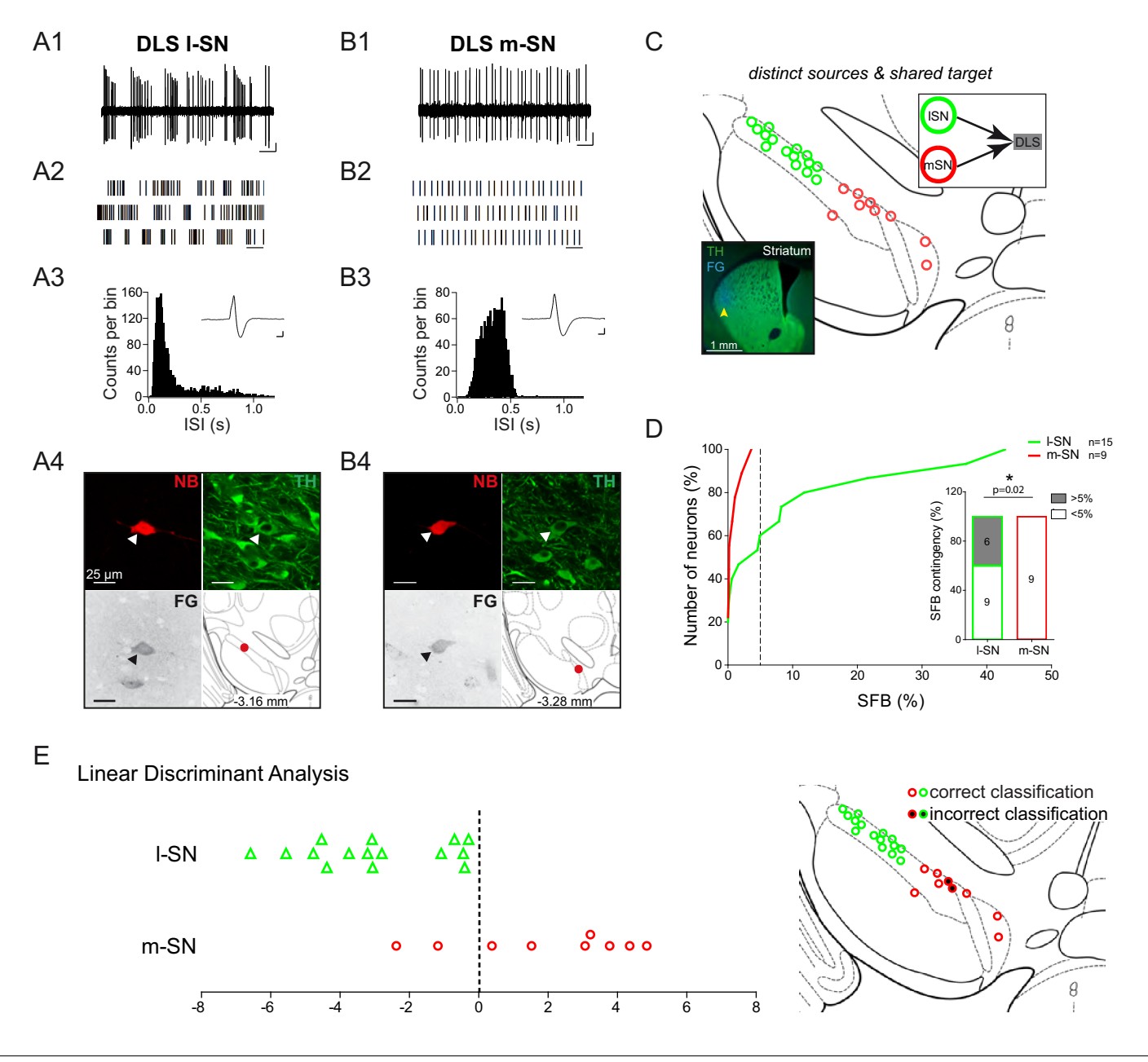

**Figure 6.** High in vivo burstiness in lateral compared to medial DLS-projecting SN DA neurons. (**A1, B1**) Spontaneous in vivo extracellular single-unit activities of a representative DLS-projecting DA neuron located in the lSN (**A1**) and a representative DLS-projecting DA neuron located in the mSN (**B1**), shown as 10 s of original recording traces (scale bar: 0.2 mV, 1 s). Note the differences in burstiness. (**A2, B2**) 30 s raster plots (scale bar: 1 s). (**A3, B3**) ISI-distributions. Note the presence of ISIs below 80 ms and 160 ms indicating bursts in A3 in contrast to B3. Inset, averaged AP waveform showing biphasic extracellular action potentials in high resolution (scale bar: 0.2 mV, 1 ms). (**A4, B4**) Confocal images of retrogradely traced, extracellularly recorded and juxtacellularly labelled DA neurons, the location of the neurons is displayed in the bottom right images. (**C**) Anatomical mapping of all extracellularly recorded and juxtacellularly labelled neurons (projected to bregma −3.16 mm; DLS-lSN in green, DLS-mSN in red). Inset, FG-injection site in DLS (FG in blue, TH in green). (**D**) Cumulative SFB distribution histograms (dotted line at SFB = 5% threshold) and bar graphs of SFB contingencies (% of neurons > and < 5% SFB) showing significant differences in burstiness. Note that no DLS-projecting DA neurons located in the mSN displayed a SFB above 5%. (**E**) Linear discriminant analysis of DLS-projecting DA neurons located either in the mSN or lSN. (Right Picture) Mapping of correctly- vs incorrectly-classified DLS-projecting DA neurons located in mSN or lSN.

DOI: https://doi.org/10.7554/eLife.48408.021

The following figure supplement is available for figure 6:

**Figure supplement 1.** Comparison of in vivo firing properties of lSN and mSN DA neurons projecting to DLS.

*Figure 6 continued on next page*

*Figure 6 continued*

DOI: https://doi.org/10.7554/eLife.48408.022

fluorogold protocol. In comparison to our previous work characterizing the in vitro properties of DA neurons with distinct axonal projections (*Lammel et al., 2008*), we found a number of unexpected results in vivo. We did not anticipate the large degree of functional diversity regarding burst firing among those SN DA neurons projecting to dorsolateral striatum. Furthermore, we were surprised to find in vivo bursting of DA neurons in the medial SN to be exclusively associated with cells projecting to lateral shell of nucleus accumbens. Finally, the difference in mean firing rates between slower-discharging DA neurons projecting to medial shell and faster-firing DA neurons projecting to lateral shell of nucleus accumbens was unexpected given their respective intrinsic excitabilities. Overall, our findings provide a refined functional definition of distinct DA projections in the intact brain.

Our study has a number of limitations. Importantly, all in vivo electrophysiological data were recorded under isoflurane anesthesia. Therefore, we cannot directly draw conclusions about behaviorally relevant activity patterns in DA subpopulations but rather identify differences in in vivo baseline excitability. We have previously shown that some of these differences are associated with altered behaviour, as in the case of burst firing in the medial SN and novelty-induced locomotion (*Schiemann et al., 2012*). In addition, we also showed that differences in in vivo firing rates between VTA DA neurons identified under isoflurane anesthesia in a mouse model of cognitive impairment (*Krabbe et al., 2015*), were recapitulated in awake behaving animals (*Duvarci et al., 2018*). However, the task-related differences in firing were only revealed in awake behaving animals (*Duvarci et al., 2018*). A second limitation is that we tried but did not succeed in getting more than an anecdotal data set of medial prefrontal projecting DA neurons. The unique mismatch between the extent of retrograde labelling and the difficulty of identifying and recording mesocortical DA neurons in vivo indicated that this DA subpopulation might either be particularly vulnerable to our labelling protocol or mostly silent under isoflurane anesthesia. The latter possibility is not unlikely given the evidence for orexinergic modulation of mesocortical DA neurons (*Del Cid-Pellitero and Garzón, 2014*). Furthermore, our study focused on rostral striatal areas not covering the caudal 'tail' region, which is a projection area of DA neurons located in the SN pars lateralis. This DA projection has been shown to be involved in the computing of aversive stimuli and stable visual memories in rodents and non-human primates (*Kim et al., 2017*; *Kim et al., 2015a*; *Matsumoto and Hikosaka, 2009*; *Menegas et al., 2018*; *Menegas et al., 2015*).

Our study has a number of specific implications for future work. Given that labelling of DA subpopulations – for identification and/or optogenetic manipulation – has become an essential tool, we believe it is crucial to control whether the labelling itself will change functional properties of the targeted neurons in vivo. In addition to the effect of conventional concentrations of fluorogold, we have previously shown that virally-mediated GFP-expression in SN DA neurons leads to robust changes in their intrinsic firing pattern (*Schiemann et al., 2012*). We currently do not know how general these perturbations of in vivo activity of DA neurons are, regarding the type of label, its concentration and post-injection timing.

Another notable result of our study is the large difference in burstiness between dorsolateral striatum projecting DA neurons in the medial compared to the lateral SN. While the underlying mechanisms need to be resolved, our findings are in line with a previous study from the Khaliq lab that found differences in in vitro excitability, including T-type calcium channel mediated rebound excitability, between calbindin-positive and calbindin-negative SN DA neurons (*Evans et al., 2017*). Also, Lerner and colleagues demonstrated larger HCN-currents in DLS-projecting SN DA neurons compared to DMS-projecting SN DA neurons, which might further enhance rebound excitability and possibly in vivo bursting (*Lerner et al., 2015*). In addition to the functional implications for nigrostriatal signaling, the enhanced burst excitability might also be of pathophysiological relevance. An influential hypothesis regarding the differential vulnerability between distinct DA subpopulations in Parkinson disease proposes that different degrees of activity-dependent calcium loading might play a crucial role for selective neurodegeneration (*Surmeier et al., 2017*). Given that the lateral SN is the most vulnerable DA region in PD (*Damier et al., 1999*; *Gibb and Lees, 1991*), we believe that the

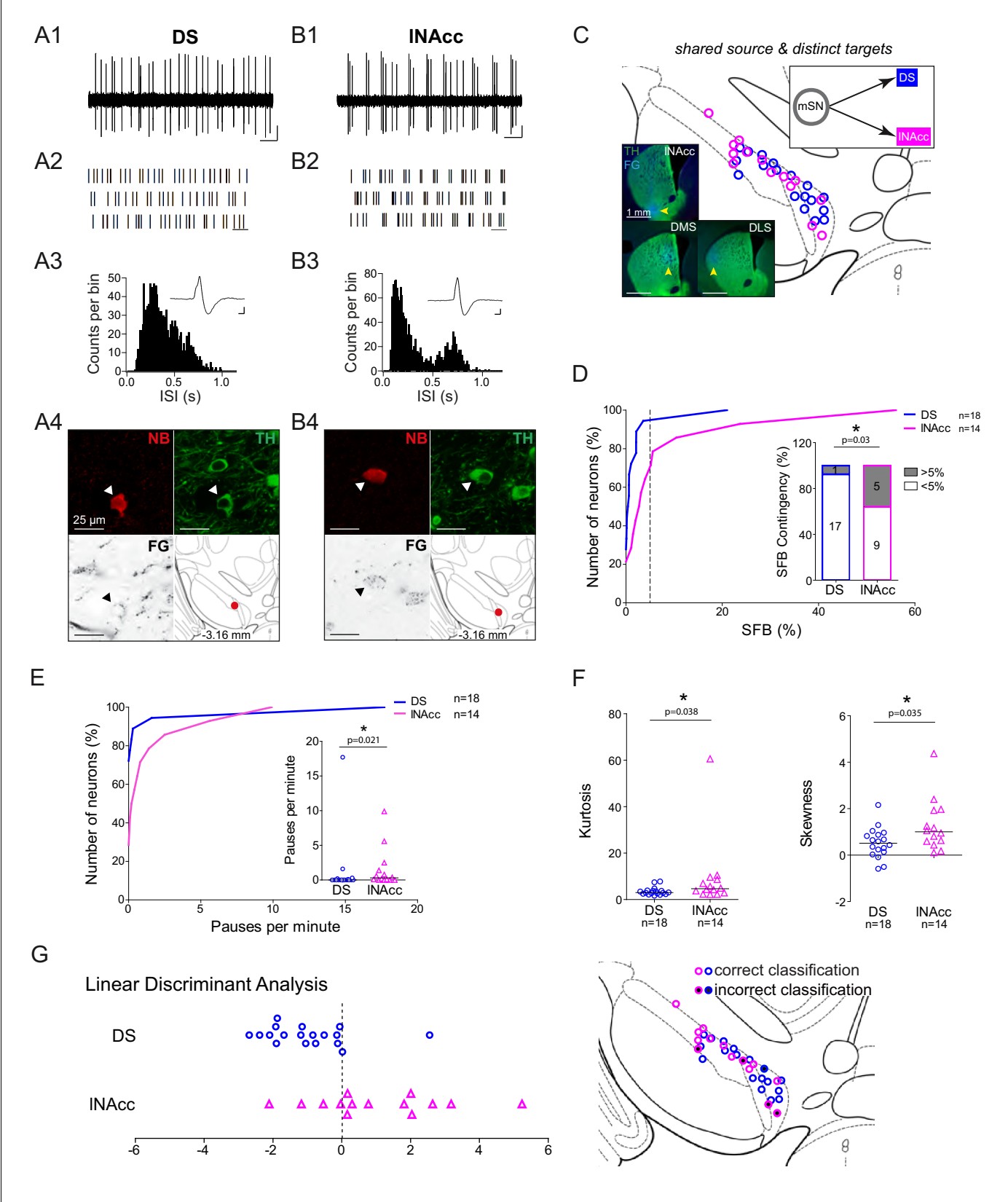

**Figure 7.** Functional segregation of in vivo electrophysiological properties mSN DA neurons with distinct axonal projections. (**A1, B1**) Spontaneous in vivo extracellular single-unit activities of a representative DS-projecting DA neuron located in the mSN (**A1**) and a representative lNAcc-projecting DA

*Figure 7 continued on next page*

*Figure 7 continued*

neuron located in the mSN (B1), shown as 10 s of original recording traces (scale bar: 0.2 mV, 1 s). Note the differences in bursting. (A2, B2) 30 s raster plots (scale bar: 1 s). (A3, B3) ISI-distributions. Note the presence of ISIs below 80 ms and 160 ms indicating bursts in B3 in contrast to A3. Inset, averaged AP waveform showing biphasic extracellular action potentials in high resolution (scale bar: 0.2 mV, 1 ms). (A4, B4) Confocal images of retrogradely traced, extracellularly recorded and juxtacellularly labelled DA neurons, the location of the neurons is displayed in the bottom right images. (C) Anatomical mapping of all retrogradely traced, extracellularly recorded and juxtacellularly labelled neurons (projected to bregma −3.16 mm; DS-lSN in blue, lNAcc-mSN in magenta). Inset, FG-injection sites in DMS, DLS and lNAcc (FG in blue, TH in green). (D) Cumulative SFB distribution histograms (dotted line at SFB = 5% threshold) and bar graphs of SFB contingencies (% of neurons > and < 5% SFB) showing significant differences in burstiness. (E) Cumulative distribution histograms and bar graphs of pauses per minute showing significant differences. (F) Scatter dot-plots showing significant differences in kurtosis and skewness of the ISI-distributions. (G) Linear discriminant analysis of DA mSN neurons projecting either to DLS/DMS or lNAcc. 70.8% of neurons were classified correctly based on skewness of the ISI-distribution, AP duration, repolarization speed and the precision of spiking ($\log(\sigma_2)$)) when randomly bisecting the data 1000 times. When the whole data set was used, 84.4% of neurons were classified correctly. (Right Picture) Mapping of correctly- vs incorrectly-classified DA mSN neurons projecting to DS or lNAcc.

DOI: https://doi.org/10.7554/eLife.48408.023

The following figure supplements are available for figure 7:

**Figure supplement 1.** DLS and DMS-projecting DA neurons located in the mSN do not exhibit significantly different in vivo firing properties.
DOI: https://doi.org/10.7554/eLife.48408.024

**Figure supplement 2.** Comparison of in vivo firing properties of DS and lNAcc-projecting DA mSN neurons.
DOI: https://doi.org/10.7554/eLife.48408.025

**Figure supplement 3.** lNAcc-projecting DA neurons located either in mSN or VTA do not exhibit significantly different in vivo firing properties.
DOI: https://doi.org/10.7554/eLife.48408.026

high in vivo burstiness of lateral SN DA neurons might be an attractive new target for protective intervention.

In contrast, we were surprised to find that mSN DA neurons projecting to dorsal striatum virtually showed no in vivo burst firing. This implies that K-ATP channel dependent burst mechanisms, which we previously identified to be selective for the medial SN (*Schiemann et al., 2012*), might be operative in lateral shell-projecting rather than dorsal striatum-projecting DA neurons in this region. The mechanism for this region-specific absence of bursting are currently unknown but might be highly relevant given that selective dysregulation in DA projections to dorsomedial striatum has been associated with schizophrenia (*Kegeles et al., 2010*; *McCutcheon et al., 2019*). While cue- and reward-triggered burst discharge has been well characterized, the function of on-going burst 'states' is less clear in the DA system. However, we and others have recently shown that DA neurons become phase-locked to prefrontal 4 Hz rhythms during working memory tasks, which might also result in rhythmic bursting (*Duvarci et al., 2018*; *Fujisawa and Buzsáki, 2011*).

Finally, our study gives further support to the idea that mesolimbic DA neurons projecting to medial versus lateral shell of accumbens constitute two parallel pathways. Recent studies highlighted the differential functional roles of these DA projections, as well as their distinct circuit affiliations (*de Jong et al., 2019*; *Lammel et al., 2012*; *Yang et al., 2018*). Our firing frequency data suggest that these two DA projections operate at different baseline excitation/inhibition (E/I) setpoints in vivo. In particular, medial shell projecting DA neurons, which displayed high intrinsic excitability associated with sustained high frequency firing in vitro (*Lammel et al., 2008*), discharged in vivo at a lower rate, which might be indicative for a higher degree of tonic net inhibition. Based on the systematic firing differences between these two mesolimbic DA subpopulations, that intermingle mainly within the VTA, it will crucial for future work in awake animals to disentangle them.

In summary, we have provided a refined source-target functional in vivo topography of midbrain DA neurons (*Figure 9*), which we believe will facilitate our understanding of dopamine neuron diversity in health and disease.

## Materials and methods

### Mice

Male C57Bl6/N mice (Charles River Laboratories; RRID: IMSR_CRL:027) were used for the study. Mice were between 11–16 weeks old, group-housed and maintained on a 12 hr light-dark cycle. All experiments and procedures involving mice were approved by the German Regierungspräsidium

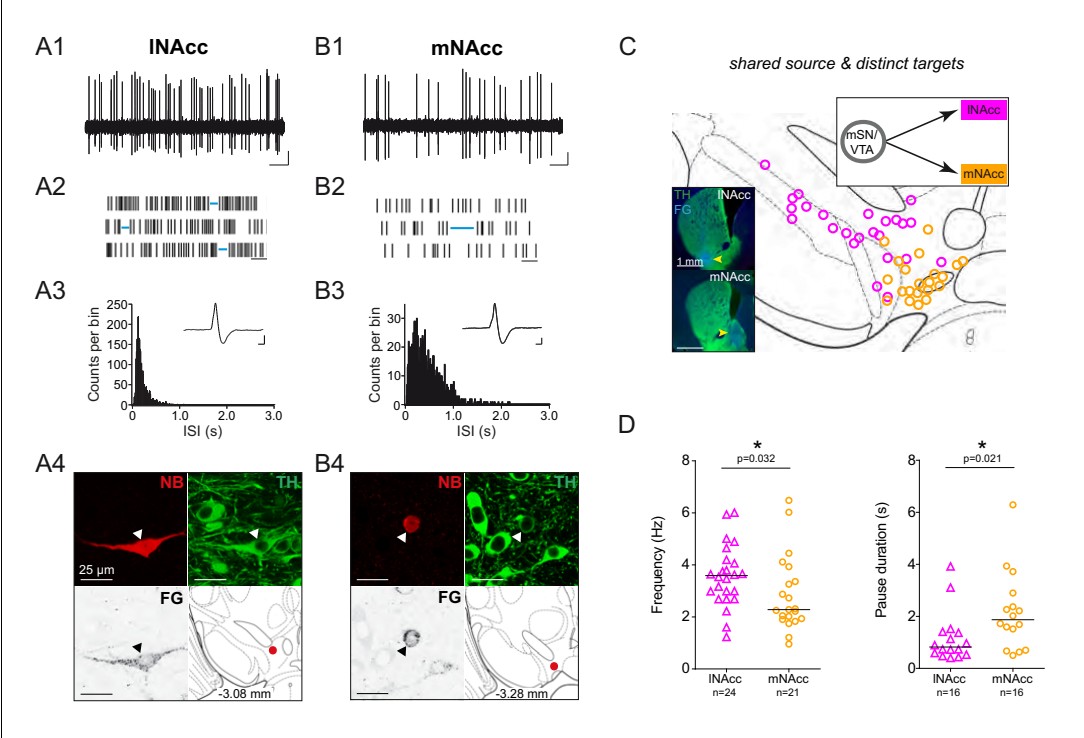

**Figure 8.** DA neurons projecting to medial or lateral shell of the nucleus accumbens display distinct in vivo firing rates. (**A1, B1**) Spontaneous in vivo extracellular single-unit activities of a representative lNAcc-projecting DA neuron located in the mSN (**A1**) and a representative mNAcc-projecting DA neuron located in the VTA (**B1**), shown as 10 s of original recording traces (scale bar: 0.2 mV, 1 s). Note the differences in firing frequency. (**A2, B2**) 30 s raster plots (scale bar: 1 s). Blue lines indicate pauses. Note the differences in pause duration. (**A3, B3**) ISI-distributions. Note the differences in overall shape and maximal ISIs. Inset, averaged AP waveform showing biphasic extracellular action potentials in high resolution (scale bar: 0.2 mV, 1 ms). (**A4, B4**) Confocal images of retrogradely traced, extracellularly recorded and juxtacellularly labelled DA neurons, the location of the neurons is displayed in the bottom right images. (**C**) Anatomical mapping of all extracellularly recorded and juxtacellularly labelled neurons (projected to bregma −3.16 mm; lNAcc in magenta, mNAcc in orange). Insert, FG-injection sites in lNAcc and mNAcc (FG in blue, TH in green). (**D**) Scatter dot-plots (line at median) showing significant differences in firing frequency (Hz) and pause duration (s).

DOI: https://doi.org/10.7554/eLife.48408.027

The following figure supplements are available for figure 8:

**Figure supplement 1.** lNAcc and mNAcc-projecting midbrain DA neurons do not exhibit significantly different in vivo firing properties.
DOI: https://doi.org/10.7554/eLife.48408.028

**Figure supplement 2.** Low success rate in identifying mPFC-projecting DA neurons.
DOI: https://doi.org/10.7554/eLife.48408.029

Darmstadt (V54-19c20/15-F40/28). In total, 108 mice were used for this study. Of these 108 mice, five mice were used for the FG dilution series (*Figure 1*), 12 were used for the mapping and anatomical quantification of projection-defined DA neuron subtypes (*Figure 4*), 12 were used for double-tracing experiments (*Figure 5*). For recording experiments, 79 mice were used. 12 of these for recordings in control mice, 17 for recordings in mice infused with FG in DLS, seven for recordings in DMS-infused mice, 18 for recordings in lNAcc-infused mice, 15 for recordings in mNAcc-infused mice and seven for recordings in mPFC-infused mice.

## Retrograde tracing

Mice were anesthetized using isoflurane (AbbVie, North Chicago, USA; induction 3.5%, maintenance 0.8–1.4% in O2, 0.35 l/min) and placed in a stereotaxic frame (Kopf, Tujunga, CA, USA). Eye cream (Vidisic, Dr Mann Pharma, Berlin, Germany) was applied on the eyes to prevent dehydration of the cornea. Lidocaine gel (Astra Zeneca, Wedel, Germany) was used as a local analgesic at the incision site. Rectal temperature (33–36°C), and respiration (1–2 Hz) were constantly monitored. Craniotomies were performed using a stereotaxic drill (0.75 mm diameter) to target the dorsal striatum (DS;

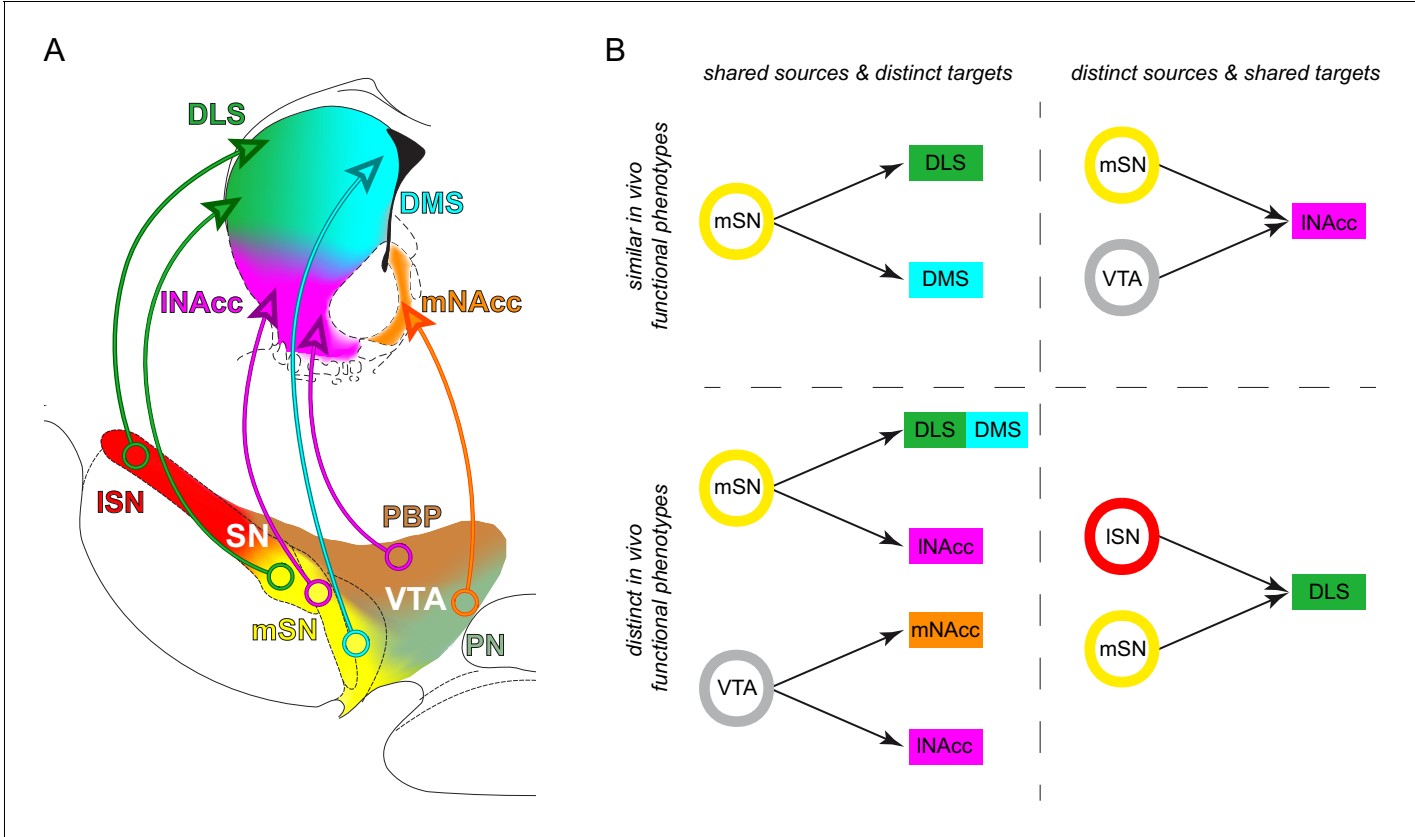

**Figure 9.** Midbrain dopamine neurons with distinct source-target topographies display functional in vivo firing differences. (**A**) Diagram of the organization of distinct midbrain DA subtypes defined by their source-target topography. Single midbrain source areas and striatal target areas are colored individually (midbrain: lSN red, mSN yellow, PBP brown, PN light green; striatum: DLS green, DMS cyan, lNAcc magenta, mNAcc orange). Distinct subtypes are illustrated with a circle in the midbrain source area and an arrow to the projection target area. (**B**) Comparisons of in vivo activities of distinct source-target topography defined DA subtypes.
DOI: https://doi.org/10.7554/eLife.48408.030

bregma: 0.86 mm, lateral: 2.0 mm, ventral: 2.8 mm), dorsolateral striatum (DLS; bregma: 0.86 mm, lateral: 2.2 mm, ventral: 2.6 mm), dorsomedial striatum (DMS; bregma: 0.74 mm, lateral: 1.4 mm, ventral: 2.6 mm), nucleus accumbens (NAcc) lateral shell (lNAcc; bregma: 0.86 mm, lateral: 1.75 mm, ventral: 4.5 mm), NAcc medial shell (mNAcc; bregma: 1.54 mm, lateral: 0.7 mm, ventral: 4.25 mm) and medial prefrontal cortex (mPFC; prelimbic nucleus (PrL; bregma: 1.98 mm, lateral: 0.3 mm, ventral: 1.5 mm) and infralimbic nucleus (IL; bregma: 1.98 mm, lateral: 0.3 mm, ventral: 2.1 mm)). Coordinates were corrected according to *Lammel et al. (2008)*.

Fluorogold (FG, Fluorochrome, Denver, USA) dissolved in artificial cerebrospinal fluid (2%, 0.2%, 0.02%, 0.002%, 0.0002%, 500 nl for DS-infusions, 1500 nl for NAcc-infusions, 2 × 1000 nl for mPFC-infusions, flow rate of 50 nl/min) and/or red beads (RB, Lumaflor, Naples, Florida, USA; 200 nl, flow rate of 100 nl/min) dissolved and diluted (1:10) in artificial cerebrospinal fluid were infused either uni- or bilaterally into the target area using a micro-pump (UMP3-1, WPI, Berlin, Germany, 10 μl nanofil syringe, 33-gauge steel needle). Post- infusion, mice were given Carprofen (Pfizer, Berlin, Germany, 4 mg/kg) subcutaneously into the nape. After a 1 week recovery period, animals were either anesthetized with Pentobarbital (Merial GmbH, 1.6 g/kg) and transcardially perfused with fixative (4% paraformaldehyde, 15% picric acid in phosphate-buffered saline, pH 7.4) for histological analysis, or used for in vivo extracellular recordings. The perfused brains were stored in paraformaldehyde overnight and changed to a 10% sucrose/0.05% NaN3 solution for long-term storage.

## In vivo electrophysiology

In vivo extracellular single-unit activities of SN and VTA neurons were recorded in untreated (Control) or post-infused (FG) mice with similar procedures as used in other studies from our lab (*Schiemann et al., 2012*). Briefly, mice were again anesthetized (isoflurane; induction 3.5%, maintenance 1–2% (>70% of recordings at 1.2%) in 0.35 l/min O2) and placed into a stereotactic frame. The craniotomies were performed as described above to target the lateral SN (bregma: −3.08 mm, lateral: 1.4 mm, ventral: 3.5–4.5 mm), medial SN (bregma: −3.08 mm, lateral: 0.9 mm, ventral: 4.2–5.0 mm), VTA (bregma: −3.08 mm, lateral: 0.75 mm, ventral: 4.0–5.0 mm). Borosilicate glass electrodes (10–25 MΩ; Harvard Apparatus, Holliston, MA, USA) were made using a horizontal puller (DMZ-Universal Puller, Zeitz, Germany) and filled with 0.5 M NaCl, 10 mM HEPES (pH 7.4) and 1.5% neurobiotin (Vector Laboratories, Burlingame, CA, USA). A micromanipulator (SM-6; Luigs and Neumann, Ratingen, Germany) was used to lower the electrodes to the recording site. The single-unit activity of each neuron was recorded for at least 10 min at a sampling rate of 12.5 kHz (for firing pattern analyses), and then for another one minute at a sampling rate of 20 kHz (for the fine analysis of AP waveforms). Signals were amplified 1000x (ELC-03M; NPI electronics, Tamm, Germany), notch- and bandpass-filtered 0.3–5000 Hz (single-pole, 6 dB/octave, DPA-2FS, NPI electronics) and recorded on a computer with an EPC-10 A/D converter (Heka, Lambrecht, Germany). Simultaneously, the signals were displayed on an analog oscilloscope and an audio monitor (HAMEG Instruments CombiScope HM1508; AUDIS-03/12M NPI electronics). Midbrain DA neurons were initially identified by their broad biphasic AP (>1.2 ms duration) and slow frequency (1–8 Hz) (*Grace and Bunney, 1984*; *Ungless et al., 2004*). AP duration was determined as the interval between the start of initial upward component and the minimum of following downward component.

## Juxtacellular labelling of single neurons

In order to identify the anatomical location and neurochemical identity of the recorded neurons, they were labelled post-recording with neurobiotin using the juxtacellular in vivo labelling technique (*Pinault, 1996*). Microiontophoretic currents were applied (1–10 nA positive current, 200 ms on/off pulse, ELC-03M, NPI Electronics) via the recording electrode in parallel to the monitoring of single-unit activity. Labelling was considered successful when the firing pattern of the neuron was modulated during current injection (i.e., increased activity during on-pulse and absence of activity in the off-pulse) and the process was stable for at least 20 s followed by the recovery of spontaneous activity. This procedure allowed for the exact mapping of the recorded DA neuron within the SN and VTA subnuclei (*Franklin and Paxinos, 2008*) using custom written scripts in Matlab 2016a (MathWorks, Natick, MA, USA; RRID: SCR_001622), combined with neurochemical identification using TH immunostaining.

## Cortical LFPs

For control of general brain state during anesthesia at different isoflurane levels, we recorded LFPs from three animals in the visual cortex with the same electrodes as those used for in vivo electrophysiology (described above). Signals were amplified 1000x (ELC-03M; NPI electronics, Tamm, Germany), notch- and bandpass-filtered 0.1–1000 Hz (single-pole, 6 dB/octave, DPA-2FS, NPI electronics) and recorded on a computer with an EPC-10 A/D converter (Heka, Lambrecht, Germany). Breathing rates were monitored. To test the depth of anesthesia, a mild nociceptive stimulus was applied by poking a hindpaw with tweezers at the end of each recording period.

## Immunohistochemistry

Following in vivo recordings, the animals were transcardially perfused as described previously. Fixed brains were sectioned into 60 µm (midbrain) or 100 µm (forebrain) coronal sections using a vibrating microtome (VT1000S, Leica). Sections were rinsed in PBS and then incubated (in blocking solution (0.2 M PBS with 10% horse serum, 0.5% Triton X-100, 0.2% BSA). Afterwards, sections were incubated in in carrier solution (room temperature, overnight) with the following primary antibodies: polyclonal guinea pig anti-tyrosine hydroxylase (TH; 1:1000; Synaptic Systems; RRID: AB_2619897), monoclonal mouse anti-TH (1:1000; Millipore; RRID: AB_827536), polyclonal rabbit anti-TH (1:1000; Synaptic Systems; RRID: AB_2619896), polyclonal rabbit anti-FG (1:1000, Fluorochrome; RRID: AB_2314408), monoclonal mouse anti-calbindin (CB; 1:1000; Swant; RRID: AB_10000347). In sequence,

sections were again washed in PBS and incubated (room temperature, overnight) with the following secondary antibodies: goat anti-guinea pig 488 (RRID: AB_142018), goat anti-guinea pig 647 (RRID: AB_2535867), goat anti-rabbit 568 (RRID: AB_143157), goat anti-rabbit 488 (RRID: AB_143165), goat anti-mouse 647 (RRID: AB_2535804), goat anti-mouse 488 (RRID: AB_2534069) (all 1:750; Life Technologies). For striatal or prefrontal sections, anti-FG antibody and subsequent secondary antibody were not used. Streptavidin AlexaFluor-568 and Streptavidin AlexaFluor-488 (both 1:1000; Invitrogen) were used for identifying neurobiotin-filled cells. Sections were then rinsed in PBS and mounted on slides with fluorescent mounting medium (Vectashield, Vector Laboratories).

## Confocal analysis and FG detection

Multilabeling fluorescent immunostainings of recorded, juxtacellularly filled and retrogradely traced neurons were analyzed using a laser-scanning microscope (Nikon Eclipse90i, Nikon GmbH). NIS-Elements C program (Nikon software; RRID: SCR_014329) was used to acquire and export images. Low magnification images of the forebrain injection sites were acquired with a 4x objective. Overview images of the midbrain were acquired with 10x or 20x objectives. High magnification images of traced and juxtacellularly labelled neurons were acquired with a 60x oil immersion objective. For FG-labelling detection, confocal settings differed between neurons that did not undergo FG immunohistochemistry (intrinsic FG signal) and neurons that did undergo FG immunohistochemistry. Identical confocal settings were chosen within each group. For semiquantitative analysis of both intrisic and immuno-FG signals in the soma of juxtacellularly labelled SN and VTA DA neurons, we determined the mean intrinsic FG signal intensities of TH-negative background regions and TH-positive regions-of-interests in the midbrain using ImageJ software (http://rsbweb.nih.gov/ij/; RRID: SCR_003070). TH-negative nuclei were subtracted from the background. For intrinsic FG signals, a neuron was identified as FG- labelled if FG signals in the ROI were higher than the mean background intensity + 2*SD of the mean background. For immuno-FG signals, neurons were identified as FG-positive if there were >5 FG-positive marks after thresholding images at mean background + 10*SD of the mean background. Also,>95% of retrogradely FG-3-labelled neurons in both the SN and the VTA were TH-positive that is dopaminergic (99.23% of DLS-projecting midbrain neurons, 99.19% of DMS-projecting midbrain neurons, 98.57% of lNAcc-projecting neurons, 96.34% of mNAcc-projecting neurons).

## Mapping of FG injection sites

For identification of local FG infusion sites, forebrain sections were analyzed. Sections with the highest intensity FG signal for each mouse were identified, selected for analysis and defined according to their distance relative to bregma and neuroanatomical landmarks using *Franklin and Paxinos (2008)*. Fluorogold infusion sites were visually identified and marked. Overlays from distinct animals are shown on corresponding stereotactic maps (*Franklin and Paxinos, 2008*).

## Mapping and anatomical quantification of projection-defined DA neuron subtypes

In order to map projection-defined DA subtypes in the midbrain, three animals in each group (DMS, DLS, lNAcc, mNAcc) were infused with FG and processed after a 1 week survival period. After TH-, FG- and Calbindin-immunohistochemistry was performed, 10x and 20x z-stack images of 60 μm midbrain sections were acquired and every $2^{nd}$ image was used for analyses. To illustrate the anatomical segregation of different projection-defined subtypes, images from 10x z-stack images were flattened into single maximum intensity projections using Image J. Sections were defined according to their distance relative to bregma and neuroanatomical landmarks using (*Franklin and Paxinos, 2008*). Sections at the same bregma were aligned using ImageJ FIJI (RRID: SCR_002285) and merged. Image brightness was adjusted relative to maximum peak, the background was subtracted (rolling ball radius = 10), a Gaussian blur filter was employed (sigma = 5), and a colored heat map was created using ImageJ plugin by Samuel Péan (http://imagejdocu.tudor.lu/doku.php?id=plugin:analysis:heatmap histogram:start). The heat map image was then overlaid on the corresponding stereotactic map figures (*Franklin and Paxinos, 2008*).

For the quantification of projection-defined DA neuronal cell populations, 20x z-stack images of 60 μm thick midbrain sections were taken. FG-positive, FG- and TH-copositive and FG-, TH- and CB-

copositive neurons were counted manually with ImageJ cell counter by Kurt De Vos (https://imagej.nih.gov/ij/plugins/cell-counter.html). For quantification, either absolute accumulated numbers of neurons (n = 3 animals) were presented or the average percentage of neurons either located in a distinct midbrain area or projecting to a distinct target area.

### Double tracing experiments

Double tracing experiments were carried out with FG and RB as described above. Three animals in each group (mNAcc and lNAcc, lNAcc and DLS, DLS and DMS) were infused and sacrificed after a 1 week survival period. After performing immunohistochemistry, 20X z-stack confocal images were taken. Cell counting was performed in the region of highest overlap, that is for lNAcc/mNAcc, cells were counted in the VTA, for lNAcc/DLS cells were counted in the mSN and for DLS/DMS, cells were counted in the mSN. We quantified cells in caudal sections (−3,64 mm), intermediate sections (−3.28 mm) and rostral sections (−3.08 mm).

### Statistical analysis

#### Spike train analyses

Spike timestamps were extracted by thresholding above noise levels with IGORPRO 6.02 (WaveMetrics, Lake Oswego, OR, USA; RRID: SCR_000325). Firing pattern properties such as mean frequency, coefficient of variation (CV) and bursting measures were analyzed using custom scripts in Matlab (R2016b). In order to estimate burstiness and intra-burst properties, we used the burst detection methods described in *Grace and Bunney (1984)*. Only cells with estimated bursts were included for the intraburst-statistics. For intraburst-frequency analysis, the average of mean intraburst-frequencies of all bursts in a spiketrain were calculated. Pauses were detected using a custom version of the Tukey outlier identification algorithm (*Tukey, 1977*). This method relies on setting a non-parametric pause threshold that is adjusted to each cell but applied across the entire spike train. The pause threshold $pthr$ is the 75% quantile $q3$ of the ISI distribution, added to a factor $w$ (set as three in this study) multiplied by the interquartile range: $pthr=q3+3*(q3-q1)$. Also, pauses after bursts are excluded as they putatively could be of non-synaptic origin.

For analysis of general firing patterns, autocorrelation histograms (ACH) were plotted using custom Matlab scripts. As described before (*Schiemann et al., 2012*), we used established criteria for classification of in vivo firing patterns based on visual inspection of autocorrelograms: single spike-oscillatory ('Pacemaker';≥3 equidistant peaks with decreasing amplitudes), single spike-irregular ('irregular';<3 peaks, increasing from zero approximating a steady state), bursty-irregular ('irregularly bursty'; narrow peak with steep increase at short ISIs) and bursty-oscillatory ('regularly bursty'; narrow peak reflecting fast intraburst ISIs followed by clear trough and repetitive broader peaks).

For a quantitative description of firing patterns, we fitted a stochastic model called Gaussian Locking to a free Oscillator (GLO; *Bingmer et al., 2011*; *Schiemann et al., 2012*) to the spike trains. The GLO mainly includes the four parameters mean period and variability of oscillation, burst size and precision of spike locking to the oscillation ($\sigma_2$). In order to fit the GLO, we first applied a multiple filter test for rate stationarity (*Messer et al., 2017*) to identify the longest sections with approximately constant firing rate. The GLO was then used to classify this longest stationary phase into single spike oscillatory, irregular single spike, etc. The GLO analysis was performed using the statistical analysis package R (www.r-project.org; RRID: SCR_001905).

For AP duration analysis, high-resolution recordings of at least 10 APs were aligned, averaged and the interval between the start of the initial upward component and the minimum of the following downward component was measured using IGORPRO. In order to analyze the relative AP trough, aligned and averaged APs of single neurons were normalized to peak amplitude = 1 mV and the trough was measured using IGORPRO.

Numerical data is represented as scatter plots with a line on the median. Categorical data is represented as stacked bar graphs. To investigate the assumption of normal distribution, we performed the single-sample Kolmogorov-Smirnov test. Because, in general, the normality assumption was rejected, comparisons were performed with the nonparametric Mann–Whitney-Test. Categorical parameters such as GLO-based firing pattern and SFB contingency were analyzed with the Chi-squared test. Statistical significance level was set to p<0.05. All data values are presented as

means ± SEM. Statistical tests were made using GraphPad Prism 6 (GraphPad Software, San Diego, CA, USA; RRID: SCR_002798).

## Linear discriminant analysis

We applied linear discriminant analysis (LDA) to determine whether different projection-defined DA neuron subpopulations could be reliably discriminated based on their firing patterns alone. In a multidimensional data set with two or more specified groups, the LDA finds the direction that optimally separates the groups. In order to identify the set of variables contributing to this discrimination, we used the following cross validation technique: For every possible subset $S_j$ of the set of all variables (i.e., the four GLO parameters, the mean spike frequency, the CV, the SFB, the skewness and kurtosis of the interspike intervals, the burst rate, the number of estimated pauses, the AP duration and the relative AP through), we performed the following analysis: we used 1000 random divisions of the data in two equally sized groups A and B. For every division, we performed LDA on A and predicted the group affiliations of B, and vice versa. The mean percentage of correct predictions in A and B, averaged across all 1000 random divisions, is called the predictive power of the model with variable set $S_j$ and denoted by $M_j$. The maximal predictive power is then given by $\max_j M_j$ and the corresponding subset $S_j$ is called the best subset.

In order to investigate statistical significance of the maximal predictive power, we used a permutation test, performing the same analysis with 1000 randomly permuted data sets, in which the two groups were assigned randomly. Statistical significance was then assessed by deriving the percentage of these analyses of permuted data sets – as specified in the results – which yielded a maximal predictive power as large as or larger than the observed. In case of statistical significance, we also give the percentage of correct categorizations for the respective subset $S_j$ when performing LDA on the whole data set.

## Acknowledgements

This study was supported by research grants to JR (NIH Grant R01DA041705, DFG CRC 1080, CRC 1193 and a Fellowship by the Gutenberg Research College, University Mainz) and to GS by the DFG Priority Program 1665 (SCHN 1370/02–1). NF is a MD/PhD candidate at TransMed, Gutenberg University Mainz. We thank Beatrice Fischer and Jasmine Sonntag for technical assistance.

## Additional information

### Funding

| Funder | Grant reference number | Author |
| --- | --- | --- |
| National Institutes of Health | R01DA041705 | Jochen Roeper |
| Deutsche Forschungsgemeinschaft | CRC 1080 | Jochen Roeper |
| Gutenberg Forschungskolleg | | Jochen Roeper |
| Deutsche Forschungsgemeinschaft | CRC 1193 | Jochen Roeper |
| Deutsche Forschungsgemeinschaft | DFG Priority Program 1665 (SCHN 1370/02-1) | Gaby Schneider |

The funders had no role in study design, data collection and interpretation, or the decision to submit the work for publication.

### Author contributions

Navid Farassat, Conceptualization, Data curation, Formal analysis, Investigation, Methodology, Writing—original draft, Writing—review and editing; Kauê Machado Costa, Conceptualization, Formal analysis, Visualization, Methodology; Strahinja Stojanovic, Data curation, Formal analysis, Investigation; Stefan Albert, Lora Kovacheva, Josef Shin, Formal analysis, Methodology; Richard Egger, Investigation; Mahalakshmi Somayaji, Methodology; Sevil Duvarci, Investigation,

Methodology; Gaby Schneider, Conceptualization, Formal analysis, Methodology; Jochen Roeper, Conceptualization, Formal analysis, Supervision, Funding acquisition, Investigation, Methodology, Writing—original draft, Project administration, Writing—review and editing

### Author ORCIDs
Kauê Machado Costa https://orcid.org/0000-0002-5562-6495
Jochen Roeper https://orcid.org/0000-0003-2145-8742

### Ethics
Animal experimentation: All experiments and procedures involving mice were approved by the German Regierungspräsidium Darmstadt (V54-19c20/15-F40/28).

### Decision letter and Author response
Decision letter https://doi.org/10.7554/eLife.48408.034
Author response https://doi.org/10.7554/eLife.48408.035

## Additional files

### Supplementary files
• Source code 1. Matlab scripts for generation of feature mapsCustom script used in order to create feature maps as in *Figure 3—figure supplement 1*, *Figure 6—figure supplement 1*, *Figure 7—figure supplement 2* and *Figure 8—figure supplement 1*.
DOI: https://doi.org/10.7554/eLife.48408.031

• Transparent reporting form
DOI: https://doi.org/10.7554/eLife.48408.032

### Data availability
All data generated or analysed during this study are included in the manuscript and supporting files.

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
