## [Decision Letter]

Thank you for submitting your article "in vivo functional diversity of midbrain dopamine neurons within identified axonal projections" for consideration by *eLife*. Your article has been reviewed by three peer reviewers, and the evaluation has been overseen by a Reviewing Editor and Gary Westbrook as the Senior Editor. The reviewers have opted to remain anonymous. The reviewers have discussed the reviews with one another and the Reviewing Editor has drafted this decision to help you prepare a revised submission.

The study by Farrasat et al., represents an advance in understanding dopamine neuron physiology by providing a detailed in vivo topography of dopamine neuron firing activity according to projection specificity. These results provide a connection between what is already known about differences in the in vitro features with in vivo activity according to projection target. A major accomplishment was to retrogradely label the recorded neurons without affecting their in vivo firing activity. The data are of high quality and the presentation is clear. This is a well-executed study and an important step in our understanding of diversity among dopamine neuron subpopulations in the intact brain.

The reviewers have raised several important comments that must be addressed. The major comments of the reviewers are summarized and restated in the essential revisions.

Essential revisions:

1) Dopamine neurons have widespread axonal projection (see Matsuda et al., 2009) that can cover extremely large striatal area (both medial and lateral). Less is known of the DA neurons projecting to the NAc and whether a single neuron can innervate both the core and the shell. Previous literature indicates that these DA neurons form significantly smaller axonal arbors (Aransay et al., 2015). Furthermore, these authors and others have distinguished DA cells projecting to medial and lateral NAc shell (de Jong et al., 2019). It is important that the authors clearly show that their DA subtype classification (based on 'single' projection sites) is real and not overlapping (in their projection sites). The reviewers propose addressing this question with for example, retrograde tracing experiments combining fluorogold and Ctb injections in their various striatal targets.

2) Comparing in vivo firing activity of neurons requires the careful control of cortical brain state activity. The lack of stability of isoflurane anaesthesia makes the comparison of neuronal activity across animals even more daunting. While the authors are not asked to redo all their recordings with the addition of an ECoG, evidences must be presented showing that the neuronal activity of DA neurons (irrespective of their projection site) is not affected when the isoflurane concentration is changing from high (deep anaesthesia) to low (lighter anaesthesia). This can easily be done by the authors. It would clarify a key concern and be a great addition to the paper.

3) It would be more helpful to evaluate the injection sites if given a sense of the variability in injection spread and location for each target site. An atlas collecting all injection spreads and locations will increase confidence in the injection target regions being separate and would be a great resource to researchers in the field.

4) While burst firing in dopamine neurons is typically associated with a stimulus, the current data report "ongoing" burst firing (both periodic and irregular). How do the authors interpret this activity? It would be helpful to discuss instances where this type of activity has been reported in studies of dopamine neurons recorded in awake behaving animals or to provide ideas/predictions involving circuit properties that may contribute to this spiking behavior (intrinsic properties contribute, but SNc neurons do not typically burst in the absence of synaptic input).

Reviewer #1:

The study by Farrasat et al., represents an advance in understanding dopamine neuron physiology by providing a detailed in vivo topography of dopamine neuron firing activity according to projection specificity. This provides a connection between what is already known about differences in in vitro features with in vivo activity according to projection target. A major accomplishment of this study was to retrogradely label the recorded neurons without affecting their in vivo firing activity. This was accomplished by using a 10-fold lower concentration of fluorogold that the authors convincingly demonstrate does not affect the in vivo firing activity of the dopamine neuron. Of interest is also the use of the Linear Discriminate Analysis, which provides for predictive power of the firing activity to predict projection target.

Only one moderate concern should be addressed. It would be more helpful to evaluate the injection sites if given a sense of the variability in injection spread and location for each target site. An atlas collecting all injection spreads and locations will increase confidence in the injection target regions being separate.

Reviewer #2:

The paper submitted by Farassat et al., described the in vivo electrophysiological properties of subclasses of dopaminergic (DA) neurons identified based on their position in the substantia nigra (SN) or their projection site to the striatum (i.e. DLS vs. DMS) and the nucleus accumbens (i.e. shell vs. core). This kind of work is labour intensive and technically challenging for multiple reasons: 1/ all the DA neurons included in this work were neurochemically identified through in vivo juxtacellular labelling combined with post-hoc immuno-detection for the tyrosine hydroxylase; 2/ the authors further identify subclasses of dopaminergic neurons based on their site of projection through the retrograde transport of fluorogold. The results are interesting and provide additional evidence that dopaminergic neurons are functionally diverse. That being said, the electrophysiological differences described by the authors between distinct target-projecting DA neurons are subtle (i.e. minor change in firing rate or bursting) and not always present (no differences between DLS vs. DMS projecting DA neurons in medial SN). Overall, the manuscript is sound and the work of good-quality; the only downside being that the results highlight only minor (or in some cases even no differences) between DA neurons with distinct axonal projections.

1) Within one 'source' (i.e. the medial SN), the authors identify distinct 'targets' DA projecting neurons (i.e. projecting to the DMS, the DLS, or the NAcc). This is achieved using a fluorogold retrograde approach but only one target can be tested at any time. This projection-based classification is then used as a criterion for electrophysiological comparison between distinct DA neurons subclasses but this approach does not allow to identify neurons with multiple projection site, for example: DMS and DLS. This has important consequences since the authors might be comparing 'presumably' distinct population of DA neurons that in fact belong to the same cluster of neurons. It seems important here to better characterize whether or not DA neurons have multiple projection targets.

2) Another consequence of this 'single' target retrograde strategy is that the electrical activity of subclasses of DA neurons are recorded in different subject, yet compared as if they were recorded from the same animal in similar condition. Unfortunately, the level of isoflurane anaesthesia can vary a lot from one animal to another, which mean that the cortical activity will change from animal to another and this will impact onto the firing properties of DA neurons (both on the level of bursting and firing rate). Considering this important aspect and the fact that the authors did not record the cortical activity, how confident can we be that the different electrical properties of DA neurons recorded in different mice are not artificially caused by extrinsic factors such as brain state, or variability across animals? It seems important to clarify this point.

Reviewer #3:

This manuscript by Farassat et al. presents a comprehensive analysis of midbrain dopamine neuron functional diversity performed in the in vivo mouse preparation. The authors combine retrograde fluorogold labeling with juxtacellular recordings to compare firing patterns in dopamine neuron subpopulations categorized according to their major projection patterns. Comparing dopamine neurons within and between midbrain regions (i.e. medial vs. lateral SNc DA neurons), they find that burst firing cells are present in all SNc (lateral and medial) and VTA subregions examined. Interestingly, however, dorsal striatum projecting cells located in the medial substantia nigra do not burst. Additionally, medial NAc projecting cells fire tonically at lower rates than lateral NAc projecting neurons. Overall, these data further support the idea that dopamine neurons are diverse in their firing patterns and presumably in their function.

This is a well-executed study. The data are of high quality and the presentation is clear. The authors should be commended for performing and including their careful control experiments testing the effects of different concentrations of fluorogold on electrophysiological properties. In addition, the anatomical identification of cells and description of source and targets is valuable and important. Unlike the authors' past heterogeneity study in brain slices (Lammel et al., 2008), the firing patterns reported in this study likely result from a mixture of intrinsic properties as well as activity of the surrounding circuit. This is important because the recordings here were all made in anaesthetized mice which may complicate the interpretation of the data. Despite this caveat, however, I believe that this study is an important step in our understanding of diversity among dopamine neuron subpopulations in the intact brain.

1) Burst firing in dopamine neurons is typically associated with a stimulus (e.g. unexpected reward, aversive stimulus, etc.). Here, however, the authors report burst firing (both periodic and irregular) that seems to be ongoing. How do the authors interpret this activity? It would be helpful to discuss instances where this type of activity has been reported in studies of dopamine neurons recorded in awake behaving animals or to provide ideas/predictions involving circuit properties that may contribute to this spiking behavior (intrinsic properties contribute, but SNc neurons do not typically burst in the absence of synaptic input).

---

## [Author Response]

Essential revisions:1) Dopamine neurons have widespread axonal projection (see Matsuda et al., 2009) that can cover extremely large striatal area (both medial and lateral). Less is known of the DA neurons projecting to the NAc and whether a single neuron can innervate both the core and the shell. Previous literature indicates that these DA neurons form significantly smaller axonal arbors (Aransay et al., 2015). Furthermore, these authors and others have distinguished DA cells projecting to medial and lateral NAc shell (de Jong et al., 2019). It is important that the authors clearly show that their DA subtype classification (based on 'single' projection sites) is real and not overlapping (in their projection sites). The reviewers propose addressing this question with for example, retrograde tracing experiments combining fluorogold and Ctb injections in their various striatal targets.

We agree that the injection of a retrograde tracer into a single striatal sub-territory does not guarantee that labelled DA neurons project exclusively to this area, in fact that is very unlikely in the light of axonal reconstructions. In the context of this study, it was important to define the percentage of DA neurons that innervated two neighboring striatal sufficiently to result in double labelling. To test for the potential overlap of distinct projection-defined subtypes in their projection site, we carried out double tracing experiments with fluorogold (FG) and red beads (RB). We focused on three pairs of neighboring striatal projection sites – 1) lNAcc and mNAcc, 2) lNAcc and DLS, 3) DLS and DMS (N=3 for each pair). Animals were sacrificed one week after double infusions. After performing immunohistochemistry with TH and FG antibodies, 20x z-stack confocal images were taken. Counting of single- and double-labelled DA neurons was performed in the relevant midbrain region of highest potential overlap (VTA for lNAcc/mNAcc, mSN for lNAcc/DLS, and DLS/DMS) across the caudo-rostral extent of the midbrain (caudal: -3,64mm; intermediate: -3.28mm; rostral section: -3.08mm).

In essence, our results demonstrated that most (85-95%) DA SN neurons were single-labelled indicating that they predominantly projected to single striatal territories with only a minor proportion of doubled labelled DA neurons. Given our analysis of more than 2500 DA neurons, this result is likely to be representative (for mouse). Given this large degree of segregation between parallel DA projections, our approach to systematically compare their in vivo electrophysiology is validated (for details please consult new Figure 5).

Figure 5A displays the respective injection sites. Panel B shows the distribution of double-labelled and single-labelled DA neurons. Less than 5% of DA neurons (52 of 1167 DA neurons) counted in the VTA are double-labelled for mNAcc (FG) and lNAcc (RB) while 11,2% of DA neurons (45 of 402 DA neurons) in the m-SN are double-labelled for lNAcc (FG) and DLS (RB) and 14% of DA neurons (120 of 859 DA neurons) in the m-SN are double-labelled for DMS (FG) and DLS (RB).

2) Comparing in vivo firing activity of neurons requires the careful control of cortical brain state activity. The lack of stability of isoflurane anaesthesia makes the comparison of neuronal activity across animals even more daunting. While the authors are not asked to redo all their recordings with the addition of an ECoG, evidences must be presented showing that the neuronal activity of DA neurons (irrespective of their projection site) is not affected when the isoflurane concentration is changing from high (deep anaesthesia) to low (lighter anaesthesia). This can easily be done by the authors. It would clarify a key concern and be a great addition to the paper.

We agree that more information on the stability of isoflurane anesthesia and its effects on cortical states is an important addition. Here, we wish to address two related but distinct aspects. First, how does DA in vivo firing under isoflurane compare to firing in awake, freely moving animals? This is important because – apart from the obvious absence of task-related activity – DA neurons under isoflurane might either lack important aspects of their awake firing properties and/or display additional features never observed in awake behaving animals. In particular, the latter case could lead to serious misinterpretations, also in the context of this study. Second, how stable – and thus comparable – is the isoflurane anesthesia across animals? In addition, how sensitive are firing patterns of identified DA subpopulations to variations of isoflurane anesthesia.

Comparison Isoflurane - awake

We provide comparative data between the basic electrophysiological in vivo properties of immunohistochemically and projection-specifically identified DA neurons in mSN/VTA of this study – performed under isoflurane anesthesia – and pharmacologically identified mSN/VTA DA neuron in awake, freely moving animals (home cage activity) taken from our recent study (Duvarci et al., 2018). Regarding the overall ranges of mean firing frequencies as well as burst firing, mSN/VTA DA neurons recorded under isoflurane anesthesia and those during awake home cage activity are similar (see Figure 4—figure supplement 4D and E). However, electrical in vivoactivity in the awake state was consistently more irregular (i.e. larger CV) compared to isoflurane anesthesia, both driven by higher burst frequencies and longer pauses. Compared to awake, this data set indicated a compression of the dynamic range of DA firing under isoflurane. For our study, this implies that DA neuron firing is not less but more stable under isoflurane and that it might be more difficult to identify significant differences between distinct DA projections regarding bursting and pauses.

Stability of isoflurane anesthesia and cortical states

As in previous in vivo studies using anesthesia (e.g. Schiemann et al., 2012), we continuously adjusted the concentration of isoflurane during anesthesia to result in a targeted spontaneous respiratory rate between 1-2 Hz (when animals were also non-responsive to painful stimuli, a legal requirement for intraoperative anesthesia). This approach was remarkably stable across animals and resulted in the fact that 72% (67/94) of in vivo recorded DA neurons in this study were recorded exactly at 1.2% isoflurane (at 350ml/min 100% O_2_) and all neurons were recorded in the range between 1-2% isoflurane. Figure 4—figure supplement 4 plots firing frequencies, SFBs and CVs of all DA neurons reported in this study against the exact isoflurane concentration during their recording. As we did not find any significant correlations between any of the firing properties and the isoflurane concentration for any of the DA projection, we can essentially rule out any relevant effect of “anesthesia instability” on our data set. Finally, the well-controlled level of anesthesia- continuously monitored and adjust to the target breathing rate was consistently associated with an electrically silent cortex, in contrast to lower isoflurane concentrations, where animals breathed faster (> 2 Hz), showed withdrawal reflexes upon painful stimulation and displayed oscillatory cortical LFP activity (see Figure 4—figure supplement 4G).

3) It would be more helpful to evaluate the injection sites if given a sense of the variability in injection spread and location for each target site. An atlas collecting all injection spreads and locations will increase confidence in the injection target regions being separate and would be a great resource to researchers in the field.

We agree with the reviewers that the mapping of striatal injection sites of the retrograde trace is beneficial. This information is now given in Figure 4—figure supplement 3 and demonstrates the high precision and low variability of our approach.

4) While burst firing in dopamine neurons is typically associated with a stimulus, the current data report "ongoing" burst firing (both periodic and irregular). How do the authors interpret this activity? It would be helpful to discuss instances where this type of activity has been reported in studies of dopamine neurons recorded in awake behaving animals or to provide ideas/predictions involving circuit properties that may contribute to this spiking behavior (intrinsic properties contribute, but SNc neurons do not typically burst in the absence of synaptic input).

We agree with the reviewers that in contrast to the well-studied”*Schultzian*” cue- and reward-induced transient high frequency (i.e. burst) firing of DA neurons, ongoing in vivo rhythmic activity of DA neurons (i.e. fast burst firing interspersed by regular firing pauses) has been previously described (Schiemann et al., 2012; Duvarci et al., 2018) but its functional significance is less understood. However, recent evidence from Fujisawa and Buszaki, 2011, and our lab demonstrated task-related rhythmic entrainment of DA neuronal firing to long-range cortical 4 Hz oscillations (Duvarci et al., 2018). Furthermore, we showed that reduced phase-looking of DA neurons during a working memory task was associated with cognitive impairment. This – yet sparse – evidence points to additional functional roles and related firing pattern of DA neurons outside the canonical reward-prediction error framework. These aspects are now included in the revised manuscript.

Reviewer #1:[…] Only one moderate concern should be addressed. It would be more helpful to evaluate the injection sites if given a sense of the variability in injection spread and location for each target site. An atlas collecting all injection spreads and locations will increase confidence in the injection target regions being separate.

See essential revision 3.

Reviewer #2:[…] 1) Within one 'source' (i.e. the medial SN), the authors identify distinct 'targets' DA projecting neurons (i.e. projecting to the DMS, the DLS, or the NAcc). This is achieved using a fluorogold retrograde approach but only one target can be tested at any time. This projection-based classification is then used as a criterion for electrophysiological comparison between distinct DA neurons subclasses but this approach does not allow to identify neurons with multiple projection site, for example: DMS and DLS. This has important consequences since the authors might be comparing 'presumably' distinct population of DA neurons that in fact belong to the same cluster of neurons. It seems important here to better characterize whether or not DA neurons have multiple projection targets.

See essential revision 1.

2) Another consequence of this 'single' target retrograde strategy is that the electrical activity of subclasses of DA neurons are recorded in different subject, yet compared as if they were recorded from the same animal in similar condition. Unfortunately, the level of isoflurane anaesthesia can vary a lot from one animal to another, which mean that the cortical activity will change from animal to another and this will impact onto the firing properties of DA neurons (both on the level of bursting and firing rate). Considering this important aspect and the fact that the authors did not record the cortical activity, how confident can we be that the different electrical properties of DA neurons recorded in different mice are not artificially caused by extrinsic factors such as brain state, or variability across animals? It seems important to clarify this point.

See essential revision 2.

Reviewer #3:[…] 1) Burst firing in dopamine neurons is typically associated with a stimulus (e.g. unexpected reward, aversive stimulus, etc.). Here, however, the authors report burst firing (both periodic and irregular) that seems to be ongoing. How do the authors interpret this activity? It would be helpful to discuss instances where this type of activity has been reported in studies of dopamine neurons recorded in awake behaving animals or to provide ideas/predictions involving circuit properties that may contribute to this spiking behavior (intrinsic properties contribute, but SNc neurons do not typically burst in the absence of synaptic input).

See essential revision 4.